# Personalized biomarkers of multiscale functional alterations in temporal lobe epilepsy

Ke Xie [1] ✉, Ella Sahlas[1], Alexander Ngo[1], Judy Chen[1], Thaera Arafat[1], Jessica Royer [1], Yigu Zhou[1], Raúl Rodríguez-Cruces [1], Arielle Dascal[1], Benoit Caldairou[1], Fatemeh Fadaie[1], Alexander Barnett [1], Samantha Audrain [1], Sara Larivière [2], Lorenzo Caciagli[3], Raluca Pana[4], Alexander G. Weil[5], Christophe Grova[6,7], Birgit Frauscher [8], Dewi V. Schrader[9], Zhiqiang Zhang [10], Luis Concha [11], Andrea Bernasconi[1,12], Neda Bernasconi[1,12] & Boris C. Bernhardt [1,12] ✉

Temporal lobe epilepsy (TLE) is the most common pharmacoresistant epilepsy in adults, yet few patients receive curative surgery due to diagnostic and prognostic uncertainty. In a multicenter cohort, we analyzed multimodal MRI and clinical data from 282 TLE patients, 298 healthy controls, and 45 disease controls. Patient-specific deviations from typical lifespan trajectories of intrinsic brain function were mapped using normative modeling. Regional functional alterations were heterogeneous but overlapped most in the mesiotemporal cortex. Connectome-based simulations revealed abnormality spread followed structural network architecture, highlighting the hippocampus as well as paralimbic and medial default-mode regions as epicenters. Multimodal integration implicated superficial white-matter microstructural alterations as a key contributor. Supervised models achieved AUCs of 0.77 for distinguishing TLE from disease controls, 0.74 for lateralizing seizure focus, and 0.64 for predicting postsurgical seizure freedom; greater contralateral temporal deviations predicted poorer outcomes. These findings support individualized functional biomarkers for precision presurgical care in focal epilepsy.

Epilepsy is one of the most common, serious neurological disorders, affecting over 60 million people worldwide. Among its various forms, pharmacoresistant epilepsy is the most severe, with temporal lobe epilepsy (TLE) being the most prevalent syndrome in adults. Although TLE is traditionally considered a surgically amenable condition[1,2], many patients face significant barriers to efficient treatment—particularly in differential diagnosis, seizure focus lateralization, and the prediction of surgical outcomes. In recent decades, magnetic resonance imaging (MRI) has emerged as a powerful tool for in vivo assessment of brain structure and function and is widely applied in the diagnosis and clinical management of TLE[3–9]. However, a growing number of studies highlight considerable inter-individual variability in the distribution and severity of cortex-wide structural and functional abnormalities in TLE[10–14], underscoring the complex and heterogenous nature of this disorder. Addressing this heterogeneity is increasingly recognized as a critical step toward elucidating underlying neurobiological mechanisms and developing robust, quantitative MRI biomarkers to support precision diagnosis and prognosis in TLE[10,11].

Quantitative MRI studies of TLE have traditionally relied on classic case-control designs, comparing the patient group to healthy controls

to identify common group-level patterns of brain abnormalities. While this method has been instrumental in uncovering hallmark features of TLE, such as mesiotemporal atrophy and hyperexcitability[6–8,15,16], it provides limited insights into the variability of underlying pathophysiological mechanisms across individuals, as it does not localize the precise anatomical loci of pathological changes in any given patient. To move toward more personalized insights, new paradigms are needed to better model and explain inter-patient variability. Recent studies using clustering techniques have identified various TLE subtypes with distinct patterns of gray and white matter changes, each associated with differential cognitive and clinical profiles[10,17–19]—reinforcing the importance of addressing disease heterogeneity. While prior work has predominantly focussed on structural alterations in TLE[10,17], there is growing recognition of distributed functional disruptions[8,9]. Nevertheless, most structural and functional studies in TLE remain limited to group-level inference, posing a significant barrier to the development of patient-specific biomarkers and the implementation of precision medicine.

In this study, we leveraged a multicenter, multimodal MRI dataset to characterize intrinsic functional imbalances in TLE across various spatial scales at the single-patient level using normative modeling. MRI-based normative modeling is an emerging family of techniques that shifts the analytical focus from group-average to intra-cohort variability, enabling the detection of patient-specific brain alterations[20–24]. This technique models the relationship between MRI features and biological variables (*e.g.*, age, sex) to establish normative percentiles of variation for each brain region in healthy populations[25–27]. Each patient can then be positioned within these normative distributions to quantify regional deviations (*i.e.*, W-score), identifying where and how an individual differs from expected norms. Unlike conventional z-normalization, which relies solely on the mean and standard deviation of healthy controls[28], normative modeling accounts for potential confounders (most notably age and sex effects) in deviation mapping. This technique, widely applied in neurological and psychiatric disorders[29–34], has proven effective in revealing individualized patterns of variations in brain structure and function, motivating its application in the context of TLE.

Here, we sought to characterize brain functional deviations in individuals with TLE across multiple spatial scales and to assess the extent to which spatially heterogeneous abnormalities converge within specific brain regions or distributed networks. To this end, we conducted one of the largest resting-state functional (rs-fMRI) studies in TLE to date, comprising 282 TLE patients, 298 healthy controls, and 45 disease controls (with extratemporal focal epilepsy). We estimated region-specific W-scores to index functional deviations from normative models at the individual level and computed the group-level overlap of extreme deviations. Next, we explored how global brain connectivity architecture constrains local functional abnormalities and identified potential disease epicenters using a structural connectome-based modeling framework. Given the known co-occurrence of extended gray and white matter pathology in TLE, we further examined the spatial correspondence between cortical functional and structural deviation maps. Finally, we evaluated the utility of patient-specific functional deviation scores for clinical translation using supervised machine learning to classify TLE patients versus disease controls, lateralize seizure onset zones in TLE, and predict postsurgical seizure freedom in surgical candidates.

## Results

### Data Samples
We analyzed 625 participants, including 282 patients with TLE (133 males; mean ± SD age = 32.3 ± 10.0 years, range: 18–64 years), 298 healthy individuals (152 males; 29.3 ± 8.0 years [18–60 years]) and 45 disease controls with extratemporal focal cortical dysplasia (FCD) (23 males; 27.2 ± 7.9 years [18–54 years]). Participants were selected from 4 independent datasets of 3 epilepsy centers: (i) Montreal Neurological Institute-Hospital (*MICA-MICs*: TLE/healthy controls/FCD = 57/100/17; *NOEL*: 72/42/28)[35], (ii) Universidad Nacional Autónoma de México (*EpiC*: 29/34)[19]; and (iii) Jinling Hospital (*Nanj*: 124/122)[36]. Details on subject inclusion are provided in the Methods. Detailed demographic and clinical information for each dataset is provided in Table 1 and Fig. 1A.

### Region-specific functional deviations in TLE
Three metrics, chosen to index brain function at 3 different spatial scales (local, regional, and global), were calculated from processed rs-fMRI time series at the node level: signal temporal variability (SV), regional homogeneity (ReHo), and node strength (NS) (Fig. 1B). Normative percentiles of variations in SV, ReHo, and NS were established using data from healthy individuals, and deviations from the norms in patients were quantified as W-scores (Fig. 1C)[25,37]. Each brain region for each patient was classified as either an extreme deviation (|W-scores| ≥ 1.96) or falling within the normative range, resulting in individualized

## Table 1 | Demographic and clinical information

| dataset (n) | group (n) | age | sex (M/F) | focus (L/R) | age at onset | disease duration | HS (%) | DRE (%) | surgery (Engel I) |
|---|---|---|---|---|---|---|---|---|---|
| *all* (625) | HC (298) | 29.3 ± 8.0 (18–60) | 152/146 | – | – | – | – | – | – |
| | TLE (282) | 32.3 ± 10.0 (18–64) | 133/149 | 147/135 | 17.8 ± 10.7 (0.3–60) | 14.4 ± 11.0 (0.8–49) | 192 (68.1%) | 214 (75.9%) | 99 (74) |
| | FCD (45) | 27.2 ± 7.9 (18–54) | 23/22 | 19/26 | 11.5 ± 6.9 (0.5–27) | 15.9 ± 10.3 (0.5–48) | – | 40 (88.9%) | 36 (25) |
| *MICs* (174) | HC (100) | 32.1 ± 7.4 (20–60) | 53/47 | – | – | – | – | – | – |
| | TLE (57) | 36.8 ± 11.3 (20–64) | 31/26 | 35/22 | 21.1 ± 13.1 (0.5–60) | 15.7 ± 11.1 (1–45) | 24 (42.1%) | 50 (87.7%) | 15 (12) |
| | FCD (17) | 26.9 ± 7.5 (19–42) | 7/10 | 6/11 | 11.5 ± 6.0 (0.7–22) | 16.0 ± 9.5 (0.5–34) | – | 12 (70.6%) | 8 (5) |
| *EpiC* (63) | HC (34) | 32.9 ± 11.3 (18–57) | 10/24 | – | – | – | – | – | – |
| | TLE (29) | 31.3 ± 11.3 (18–58) | 9/20 | 18/11 | 15.0 ± 10.7 (1–40) | 16.0 ± 13.1 (1–49) | 17 (58.6%) | 29 (100%) | – |
| *Nanj* (246) | HC (122) | 25.3 ± 5.4 (19–40) | 65/57 | – | – | – | – | – | – |
| | TLE (124) | 28.3 ± 7.4 (18–48) | 66/58 | 59/65 | 16.8 ± 9.0 (0.3–43) | 11.4 ± 8.7 (0.8–37) | 113 (91.1%) | 63 (50.8%) | 35 (26) |
| *NOEL* (142) | HC (42) | 31.5 ± 8.0 (20–53) | 24/18 | – | – | – | – | – | – |
| | TLE (72) | 36.0 ± 9.7 (19–61) | 27/45 | 35/37 | 17.7 ± 11.2 (0.6–51) | 18.0 ± 12.2 (1–45) | 38 (52.8%) | 72 (100%) | 49 (36) |
| | FCD (28) | 27.4 ± 8.3 (18–54) | 16/12 | 13/15 | 11.6 ± 7.5 (0.5–27) | 15.8 ± 10.9 (3–48) | – | 28 (100%) | 28 (20) |

Age, age at seizure onset, and disease duration are presented as mean ± SD years. n indicates the number of samples in each group. Abbreviation: M = male; F = female; L = left; R = right; HS = hippocampal sclerosis; DRE = drug-resistant epilepsy; HC = healthy controls; TLE = temporal lobe epilepsy; FCD = focal cortical dysplasia.

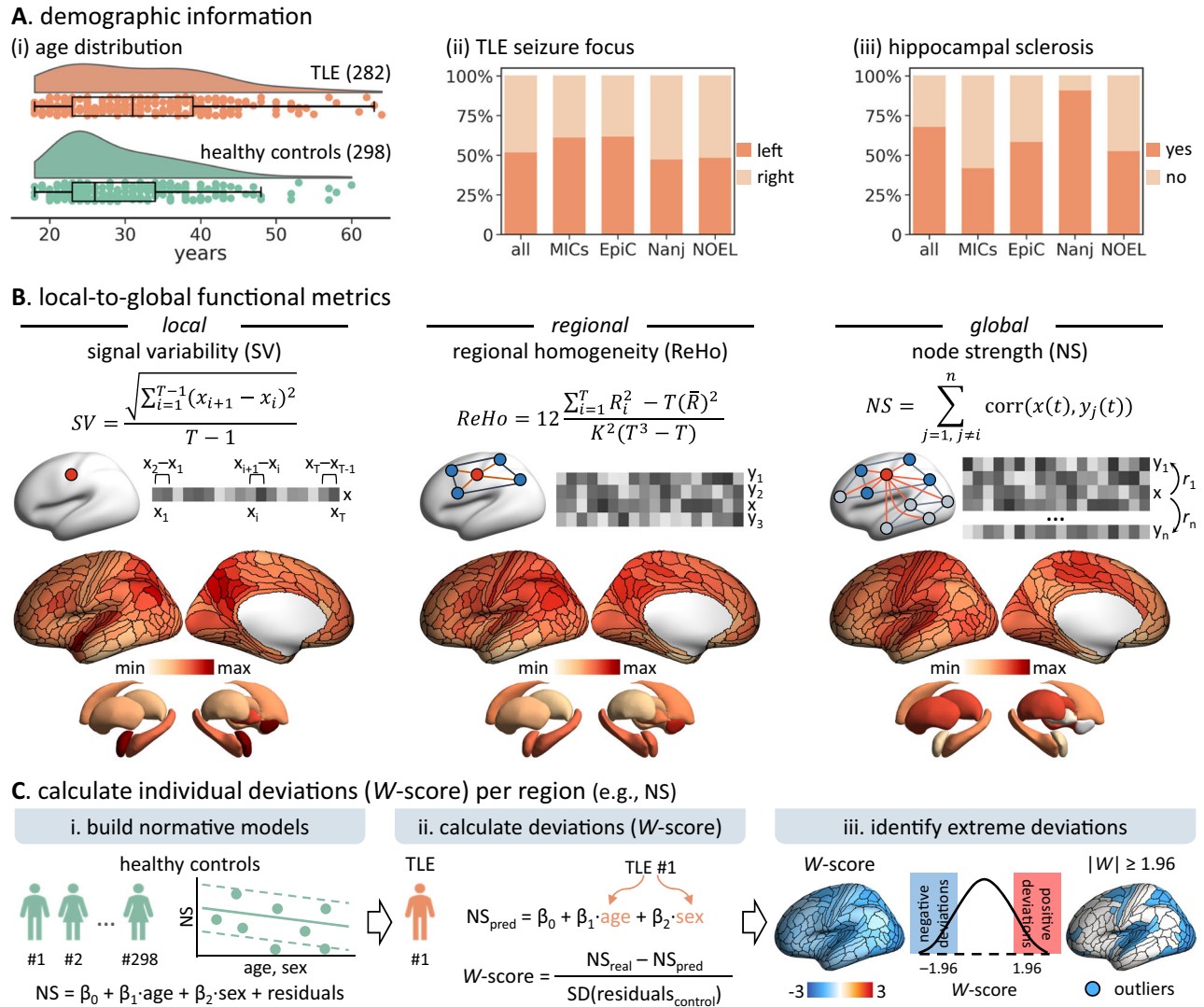

**Fig. 1 | Overview of participants and analysis pipeline. A** Demographic information of the healthy control (n = 298) and TLE groups (n = 282). *i* Age distributions. Boxplots show the median line, 25th and 75th percentiles (lower and upper bounds), and minima and maxima 1.5 interquartile range whiskers. Dots represent individual participants. *ii* Proportion of TLE patients with left- or right-sided seizure focus in each dataset. *iii* Proportion of TLE patients with or without ipsilateral hippocampal sclerosis in each dataset. **B** Overview of methodology for calculating functional metrics from resting-state functional MRI (rs-fMRI) time series in each cortical region and subcortical structure: signal temporal variability, regional homogeneity, and node strength. **C** Schematic of individual functional deviation (*W*-score) estimation, with an example here for node strength (NS). *i* Building normative models in healthy individuals. Age- and sex-related variations in each rs-fMRI metric are modeled using a linear regression model in the healthy control group, yielding beta maps for intercept ($\beta_0$), age ($\beta_1$), sex ($\beta_2$), and standard deviation of residuals for each brain region. *ii* Estimating deviations in each patient. Predicted value (*e.g.*, $NS_{pred}$) for a given patient's age and sex is calculated as $\beta_0 + \beta_1 \times age + \beta_2 \times sex$. *W*-scores are defined as the normalized deviation of the observed values from the predicted values. *iii* Identifying extreme deviations. Brain regions with *W*-scores exceeding ±1.96 (*i.e.*, | *W*-scores | ≥ 1.96) are classified as showing extreme deviations, corresponding to the upper and lower 2.5% of the normative distribution. Abbreviation: n = sample size; SD = standard deviation. Source data are provided as a Source Data file.

deviation maps[29–31]. Compared to healthy controls, individuals with TLE exhibited extensive yet variable deviations. Across the neocortex, the median proportion of extreme deviations in the TLE group was 5.3% for SV (range = 1.1%–12.4%), 7.5% for ReHo (2.1%–16.0%), and 6.4% for NS (2.1%–11.4%) (Fig. 2A, Supplementary Fig. 1). As for the subcortex, the median proportion of extreme deviations was 6.0% for SV (range = 1.8%–9.2%), 8.9% for ReHo (4.6%–13.5%), and 3.7% for NS (1.8%–6.7%). When stratified by lobe, SV and ReHo deviations were prominently located in the temporal, parietal and occipital lobes, while NS deviations were more prominent in the parietal and insular lobes, with higher prevalence ipsilateral to the seizure focus. To further explore the influence of connection distance (*i.e.*, Euclidean distance) on NS deviations, we computed short-range NS (≤ 75 mm) and long-range (> 75 mm) NS following previous work[38–40]. TLE patients showed

comparable deviation patterns for both short-range NS (neocortex: median = 6.7%, range = 2.1%–11.7%; short-range versus full NS: *rho* = 0.54, $P_{spin} < 0.001$) and long-range NS (neocortex: median = 6.4%, range = 2.1%–12.1%; long-range versus full NS: *rho* = 0.92, $P_{spin} < 0.001$; Supplementary Fig. 2). Notably, short-range NS deviations were more pronounced in the bilateral lateral prefrontal and inferior temporal cortices, while long-range NS deviations were greater in the ipsilateral insular cortex.

Next, aggregating the three metrics using a multivariate Mahalanobis distance revealed a diffuse spatial pattern of functional deviations across the brain in the TLE group (neocortex: median = 5.3% [1.8%–12.4%], subcortex: median = 5.9 [1.8%–9.9%], Fig. 2B), with the highest overlap observed in the ipsilateral mesiotemporal lobe (11.4%; Supplementary Fig. 3), putamen (9.9%), and hippocampus (7.1%). These

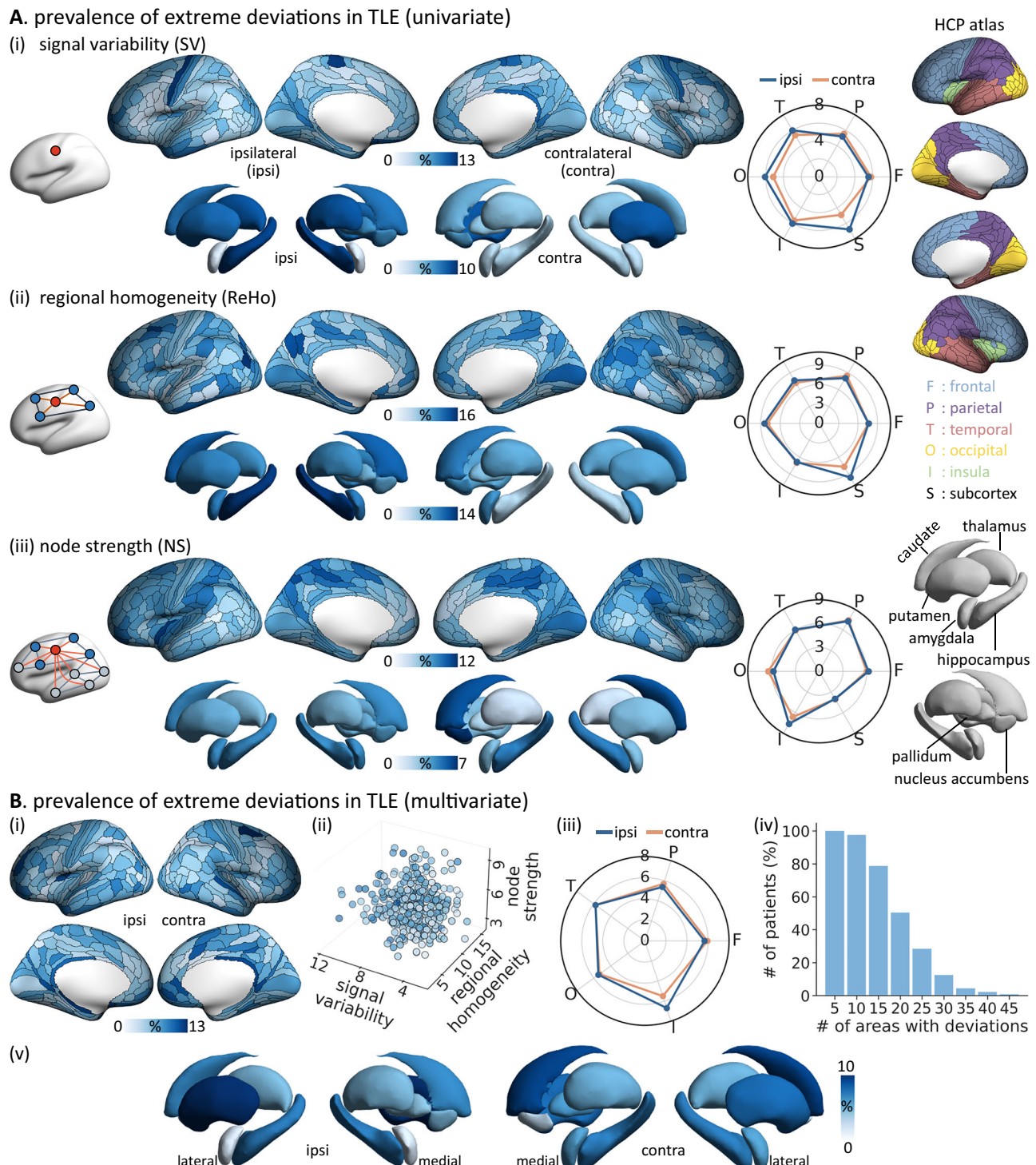

**Fig. 2 | Region-specific prevalence of extreme functional deviations in TLE patients. A** Proportion of TLE patients with extreme deviations ($|W\text{-score}| \geq 1.96$) in each cortical region and subcortical structure for (**i**) signal variability (SV), (**ii**) regional homogeneity (ReHo), and (**iii**) node strength (NS). Spider plots show the mean proportion of extreme deviations in each cortical lobe defined on the HCPMMP1.0 atlas and in the subcortex. **B i** Proportion of TLE patients with multivariate $W$-scores (aggregating SV, ReHo and NS) exceeding ±1.96. **ii** Region-specific deviation prevalence distribution. Each dot represents an individual brain region, with its position indicating the prevalence values of the three metrics and its color denoting the composite deviation prevalence. **iii** Mean proportion of extreme deviations per lobe. **iv** Distribution of the number of extreme deviations per patient. **v** Proportion of TLE patients with extreme deviations in each subcortical structure. Abbreviation: ipsi = ipsilateral; contra = contralateral; F = frontal; P = parietal; T = temporal; O = occipital; I = insula; S = subcortex. Source data are provided as a Source Data file.

findings suggest that extensive functional changes are a hallmark of TLE (97% of TLE patients had extreme deviations in at least 10 brain regions), while the specific pattern of affected regions varied markedly across individuals. We further confirmed the consistency of these findings across datasets by estimating TLE-related functional deviations in each dataset independently and correlating them with the main finding in Fig. 2B (spatial similarity: neocortex, $rho = 0.24–0.60$, $P_{spin} < 0.001$; subcortex, $rho = 0.39–0.75$, $P_{shuf} = 0.002–0.080$; Supplementary Fig. 4).

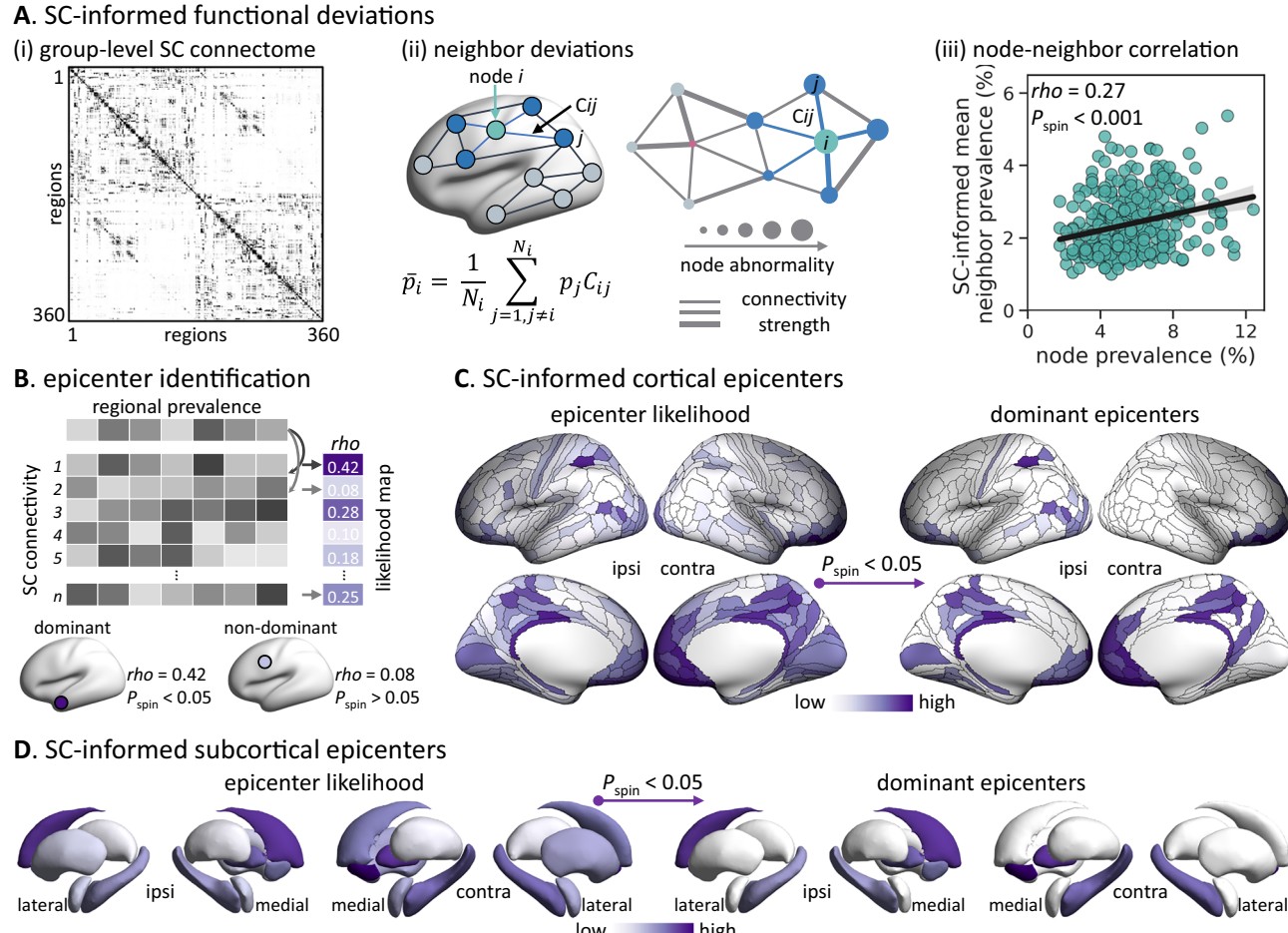

**Fig. 3 | Network-based spreading of regional functional deviations. A i** Group-level structural connectivity (SC) matrix from diffusion MRI of 100 unrelated healthy individuals. **ii** Schematic of functional deviations of a node ($p_i$) and its neighbors ($\bar{p}_i$). If the regional deviation depends on SC network organization, regions connected to highly abnormal neighbors (*i.e.*, high prevalence) will be more likely to be affected, whereas regions connected to healthy neighbors (*i.e.*, low prevalence) will be less likely to be affected. **iii** Functional deviation of each node (x-axis) versus the mean deviation of its structurally connected neighbors (y-axis). Dots are brain regions; the line shows Spearman rank fit with a 95% CI (gray band).

Correlation significance (*i.e.*, $P_{spin}$) is assessed using spin permutation tests (5000 iterations; one-sided). **B** Schematic of disease epicenter identification. A region whose SC profile spatially strongly relates to the TLE-related functional deviation map in Fig. 2B is considered as a disease "epicenter". Epicenter likelihood is defined as the Spearman correlation coefficient between two spatial maps. **C, D** SC-informed epicenter likelihood map. Statistical significance of the likelihood is determined using spin permutation tests (5000 iterations; $P_{spin} < 0.05$ against the null models, one-sided). Abbreviation: ipsi = ipsilateral; contra = contralateral. Source data are provided as a Source Data file.

Notably, the subgroup of TLE patients with hippocampal sclerosis (HS) showed more extreme deviations in the ipsilateral ($t = 2.61$, $P_{FDR} = 0.024$) and contralateral ($t = 2.75$, $P_{FDR} = 0.024$) insula cortex as well as trends in the ipsilateral thalamus ($\chi^2 = 3.68$, $P_{uncorr} = 0.055$) and pallidum ($\chi^2 = 3.21$, $P_{uncorr} = 0.073$) compared to those without HS (Supplementary Fig. 5).

### Network constraints on regional functional deviations

Since brain regions connected to others with high local vulnerability have great disease exposure, we explored the extent to which the spatial pattern of functional changes in TLE is reflected by the white matter connections. Specifically, we tested whether the structural connectivity profile of a given area $i$ could predict the functional deviation score of areas structurally connected to it[9,41]. For each brain region, we calculated the mean deviation prevalence of its structurally connected neighbors, weighted by white matter connectivity strength estimated through diffusion MRI streamline tractography (Fig. 3A). To ensure that connectivity estimates reflected typical connectomes, a group-level structural connectivity matrix was generated across 100 unrelated healthy young adults from the Human Connectome Project

(HCP)[42]. Statistical significance was assessed with spin permutation tests to account for spatial autocorrelation. We observed a positive correlation between regional deviation scores and the average deviation of their structurally connected neighbors ($rho = 0.27$, $P_{spin} < 0.001$), suggesting that the spatial patterning of TLE-related functional alterations is constrained by inter-regional structural connectivity.

Moreover, we identified disease epicenters as brain regions whose structural connectivity profiles closely resembled the spatial pattern of TLE-related functional alterations. To this end, we computed spatial correlations between each region's structural connectivity profile and the functional deviation map in the TLE group. Accordingly, the higher the correlation, the more likely the region represented a disease epicenter (Fig. 3B). Statistical significance was assessed using non-parametric permutation tests (5000 iterations), and brain regions with significant correlations ($P_{spin} < 0.05$ against null models) were identified as potential epicenters[6]. Top-ranked neocortical epicenters included the ipsilateral mesiotemporal cortex and bilateral medial prefrontal and parietal cortices (Fig. 3C). Among subcortical and mesiotemporal structures, the bilateral hippocampi, pallidum, and

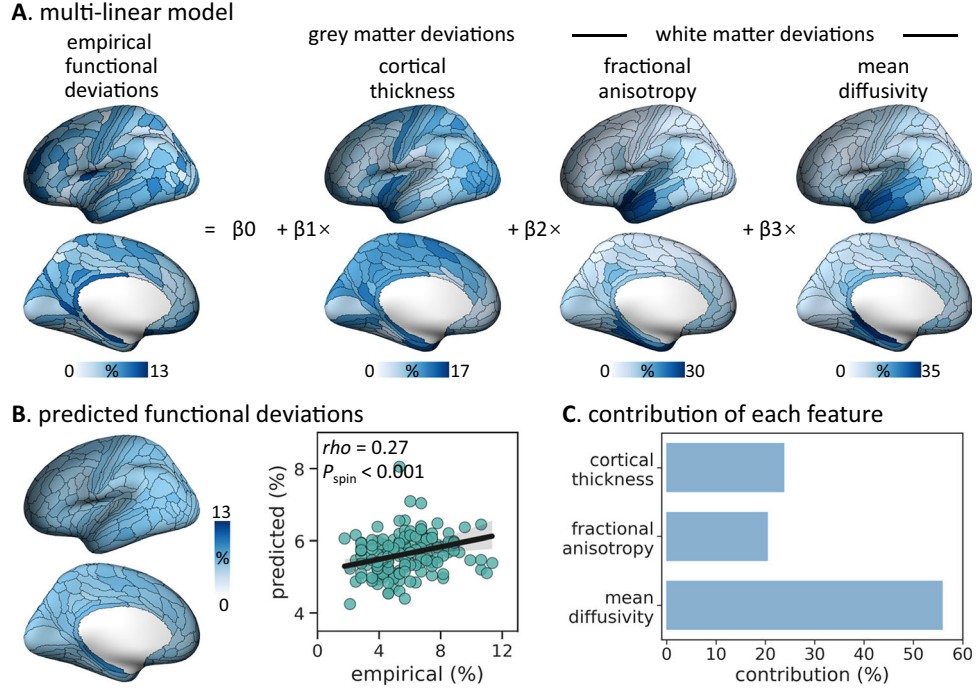

**Fig. 4 | Spatial associations between brain structural and functional deviations. A** Multilinear regression model is used to predict regional functional deviation prevalence from deviation prevalence maps of cortical thickness (CT), fractional anisotropy (FA), and mean diffusivity (MD). **B** Empirical (x-axis) versus predicted (y-axis) functional deviation prevalence maps. Dots are brain regions; the line shows Spearman's rank fit with a 95% CI (gray band). Correlation significance (*i.e.*, $P_{spin}$) is assessed using spin permutation tests (5000 iterations; one-sided). **C** Dominance analysis quantifies the contribution of each feature (CT, FA and MD). Source data are provided as a Source Data file.

nucleus accumbens, along with the ipsilateral caudate, emerged as dominant epicenters (Fig. 3D).

## Associations with structural deviations

We next examined the association between cortical functional and structural alterations in TLE using MRI markers of whole-brain structural compromise commonly employed in prior studies of this condition[8,16,43]. Cortical thickness (CT) was estimated as the Euclidean distance between corresponding vertices on the pial surface and the gray-white boundary, derived from processed T1-weighted MRI. Given that the superficial white matter (SWM) beneath the cortical interface harbors terminations of both short-range U-fibers and long-range tracts[44], it serves as an ideal substrate for integrative studies of cortical function and microstructure. We extracted SWM features—fractional anisotropy (FA) and mean diffusivity (MD)—from diffusion MRI as surrogates of fiber architecture and tissue microstructure. Analogy to brain functional metrics, individualized structural deviations (*i.e.*, CT, FA, and MD) were quantified by positioning each TLE patient's data onto normative percentile charts derived from healthy controls. Across the cortex, structural deviations were most pronounced in the temporal lobe (mean CT/FA/MD = 7.5%/14.7%/16.6%), followed by the insular cortex (8.2%/9.6%/10.6%) and, to a lesser extent, the parietal lobe (7.8%/7.6%/10.1%; Fig. 4A). To assess the relationship between structural and functional abnormalities, we employed a multilinear regression model to predict the cortex-wide prevalence of functional deviations based on the prevalence of structural deviations. We observed a positive correlation between empirical and predicted functional deviation patterns, above and beyond the effect of spatial autocorrelation (*rho* = 0.27, $P_{spin} < 0.001$; Fig. 4B). Dominance analysis further revealed that MD contributed most substantially to the model fit (56%), with modest contributions from CT (24%) and FA (20%) (Fig. 4C).

## Clinical utility of functional deviations

**TLE versus non-TLE classification.** Supervised machine learning models trained on individualized functional deviation metrics achieved an AUC of 0.77 (0.63–0.85) in discriminating TLE patients from non-TLE patients (*i.e.*, FCD-related extratemporal focal epilepsy), which significantly exceeded chance levels ($P_{perm} < 0.001$; Fig. 5A).

**Seizure focus lateralization.** Patients with left TLE could be distinguished from those with right TLE with an AUC of 0.74 (0.65–0.80) based on individualized functional deviation scores ($P_{perm} < 0.001$; Fig. 5B).

**Postsurgical seizure outcome prediction.** We examined whether inter-individual differences in preoperative functional deviations were associated with postsurgical seizure freedom in TLE patients who underwent resective surgery (n = 99). A positive spatial correlation was observed between the prevalence map of extreme deviations and the overlap map of surgical cavities (derived from pre- and post-surgical T1-weighted MRIs; mean ± SD = 39% ± 37%, range = 3%–100%) across regions (*rho* = 0.19, $P_{spin} = 0.073$; Supplementary Fig. 6), suggesting a higher likelihood of functional anomalies in to-be-resected brain regions. Moreover, in patients with postsurgical seizure recurrence, we found greater numbers of extreme deviations in the contralateral temporal lobe (*t* = 7.03, $P_{FDR} < 0.001$) and hippocampus (*t* = 1.44, $P_{uncorr} = 0.081$) compared to the ipsilateral hemisphere (Fig. 5C). In contrast, no such a significant difference was found in seizure-free patients (ipsilateral versus contralateral: temporal lobe, *t* = 0.48, $P_{FDR} = 0.407$; hippocampus, $P_{uncorr} = 0.500$). There were no significant interhemispheric differences in other lobes for either subgroup (ipsilateral versus contralateral: neocortex, $P_{FDR} > 0.084$; subcortex, $P_{uncorr} > 0.163$). Finally, supervised machine learning model trained on functional deviation scores discriminated seizure-free versus non-seizure-free patients with an AUC of 0.64 (0.51–0.73; $P_{perm} = 0.020$).

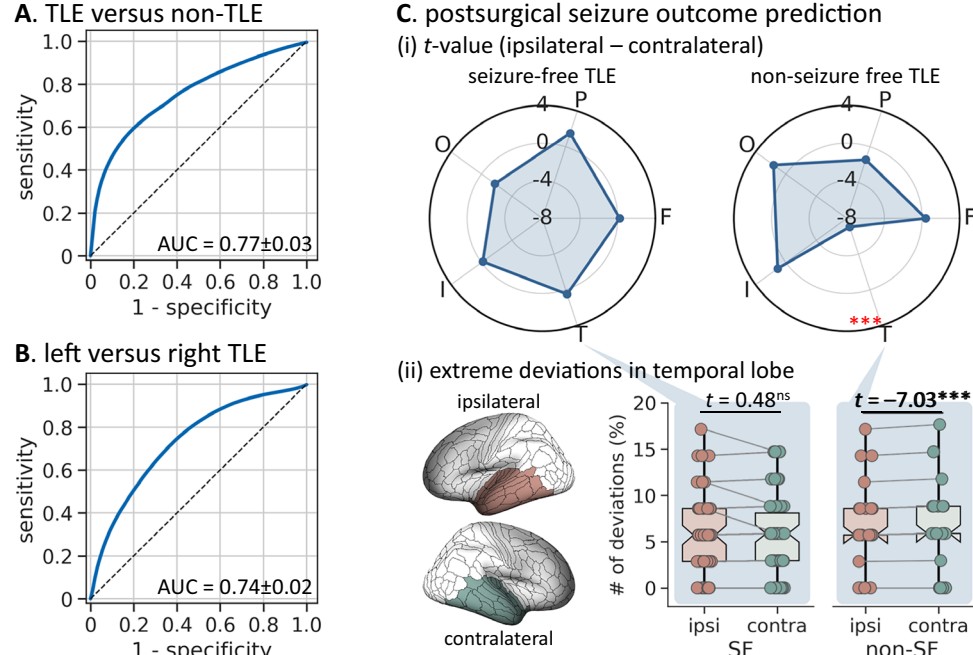

**Fig. 5 | Clinical utility of individual functional deviations.** Receiver operating curves showing the accuracy of functional deviations in (**A**) classifying TLE patients ($n = 129$) versus extratemporal FCD patients ($n = 45$), and (**B**) classifying left TLE ($n = 147$) versus right TLE ($n = 135$) patients. **C** Interhemispheric differences in the number of extreme deviations in seizure-free (SF, $n = 74$) and non-seizure-free (non-SF, $n = 25$) TLE patients. ***i*** T-values (paired *t*-test) showing the differences in the number of extreme deviations between the ipsilateral (ipsi) and contralateral (contra) hemispheres in SF and non-SF TLE patients in each cortical lobe, respectively. Positive/negative *t*-values indicate more/fewer extreme deviations in the ipsilateral hemisphere compared to the contralateral hemisphere. ***ii*** Patient-specific proportions of extreme deviations in the ipsilateral (red) and contralateral (green) temporal lobes for SF and non-SF TLE patients. Asterisks denote significance at $P < 0.001$ from one-sided paired *t*-tests after false discovery rate correction. Boxplots show the median line, 25th and 75th percentiles (lower and upper bounds), and minima and maxima, 1.5 interquartile range whiskers. Dots represent individual participants. Abbreviation: F = frontal; P = parietal; O = occipital; I = insula; T = temporal. Source data are provided as a Source Data file.

**Associations with clinical characteristics.** A positive correlation was observed between disease duration and the number of extreme deviations in the TLE cohort ($r = 0.09$, $P = 0.074$), indicating more diffuse functional deviations in patients with long-standing TLE. No significant correlations were found between the number of extreme deviations and age at seizure onset or the number of antiseizure medications ($P > 0.214$). However, patients with a history of focal to bilateral tonic-clonic seizures exhibited significantly more deviations than those without focal to bilateral tonic-clonic seizures (neocortex: $t = 1.73$, $P = 0.042$; ipsilateral caudate: $\chi^2 = 3.84$, $P_{uncorr} = 0.050$).

### Sensitivity analyses
Several sensitivity analyses confirmed the robustness of our main findings.

**Left and Right TLE.** We recomputed the functional deviation prevalence at the group level separately in left and right TLE patients. Overall, findings were spatially similar (neocortex: $rho = 0.34$, $P_{spin} < 0.001$; subcortex: $rho = 0.58$, $P_{shuf} < 0.05$; Supplementary Fig. 7); however, left TLE patients additionally had more extreme deviations in the ipsilateral temporal lobe, ipsilateral caudate and putamen, and contralateral amygdala and pallidum, while right TLE patients showed more deviations in the bilateral parietal lobe, ipsilateral pallidum, and contralateral putamen and thalamus.

**Head motion and global mean signal.** The spatial distribution of functional deviations in TLE patients was virtually identical when additionally controlling for individual's head motion during rs-fMRI scans (neocortex: median = 5.3% [1.8%–12.8%], $rho = 0.97$, $P_{spin} < 0.001$; subcortex: $rho = 0.98$, $P_{shuf} < 0.001$), and global mean signal (neocortex: median = 5.3% [1.8%–12.1%], $rho = 0.71$, $P_{spin} < 0.001$; subcortex: $rho = 0.57$, $P_{shuf} < 0.010$; Supplementary Fig. 8).

## Discussion
In this study, we quantified and visualized interindividual differences in patterns of cortical functional alterations in a large, multicentric dataset of individuals with TLE alongside with healthy controls and a disease control group with FCD-related extratemporal epilepsy. Although brain functional deviations were most prominent in the temporal lobe, there remained substantial variability across patients, indicating that TLE disrupts large-scale brain functional organization in a non-uniform manner even within its typical temporolimbic focus. The spatially distributed pattern of functional alterations nevertheless aligned with the brain's structural connectome architecture, suggesting a pivotal role of white matter tracts as conduits for the propagation of functional disturbances[9,43,45]. The hippocampus, paralimbic and medial default mode cortices emerged as epicenters of macroscale dysfunction, consistent with the presumed pathological substrate of TLE. These functional changes were primarily reflective of concurrent white matter microstructural compromise, followed by cortical thinning. Machine learning models informed by individual functional deviations successfully identified diagnostic categories and seizure focus laterality with medium-to-high accuracy and predicted postsurgical seizure outcome with moderate accuracy. Collectively, our findings show that functional alterations in TLE patients are common yet spatially heterogeneous, demonstrating the promise of patient-specific brain deviation mapping for clinical diagnosis and prognosis.

Studying multicentric rs-fMRI data from four independent datasets, we employed a normative modeling approach to quantify individual-patient deviations in brain function and generate region-specific prevalence maps of extreme deviations in TLE. Compared to

healthy controls, TLE patients presented with extensive alterations in SV, ReHo and NS, commonly affecting temporolimbic and parietal cortices, albeit with relatively low proportions of significant deviations across patients (up to 16%). Despite this variability, 97% of TLE patients had extreme deviations in at least 10 brain areas. In particular, the most evident deviations were located in the ipsilateral mesial temporal structures. These are areas known to be implicated in the pathogenesis of TLE and are responsible for its clinical manifestation[46,47]. While alterations in SV, ReHo and NS are indeed core pathopsychological features evident in most individuals with TLE[7,8,43,48–50], our findings suggest that the specific cortical regions affected by these changes differ markedly between patients. TLE impacts individuals in such a heterogenous manner that group-consensus maps of TLE pathology derived from canonical disease-control inferential paradigms may not fully capture the spectrum of individual disease manifestations[20,21,51]. While the subgroup of patients with HS-TLE had an overall pattern of functional alterations that was broadly consistent with the full cohort, they exhibited increased involvement of limbic and subcortical structures in the ipsilateral hemisphere (particularly the insula, thalamus and pallidum). These results are consistent with previous MRI-based studies that have reported widespread gray and white matter abnormalities[4,52–54], as well as a more pronounced alteration in hippocampal functional connectivity[55], network controllability[56], and excitation/inhibition imbalance in HS-TLE[9]. Additionally, they align with *post-mortem* work showing that extrahippocampal pathology in HS-TLE often extends beyond mesiotemporal regions into temporopolar, frontopolar and orbitofrontal cortices[57]. Altogether, these results support the view that HS-TLE is associated with a more severe disruption of large-scale brain networks. While our main analyses focused on the full cohort to provide a broad characterization of functional alterations across the spectrum of TLE, our subgroup results underscore the modulation of network alterations by the degree of pathological alterations in the mesiotemporal disease epicenter. Encouragingly, we found that patterns of functional brain deviations were highly consistent across sites and robust to the inclusion of global signal regression as an additional denoising step. A key strength of our study lies in the harmonization of preprocessing and analytic workflows across centers, alongside statistical adjustment for site effects. This step enhances the reliability and generalizability of our multi-site findings by minimizing nonbiological variability arising from scanner and acquisition protocol differences[58].

The human brain is an intricate network of functionally specialized brain regions. Structural connections between these regions allow for the coordination of functional interactions and the transport of trophic signals[59–62]. They can also act as conduits for the spread of pathology, such that focal perturbations can propagate to impact distributed systems[41,63,64]. Our findings align with this theory, demonstrating that functional deviations in TLE are more tightly coupled between regions with strong structural connectivity. While our study is cross-sectional in nature, we posit that connectome architecture exerts a strong influence on the spread of cortical functional damage in TLE. Such functional alterations may arise from a transsynaptic network process[63,64], in which dysregulations of the pathophysiological core may trigger atypical activity in adjacent brain structures that, over time, may result in widespread brain dysfunction affecting multiple systems. Prior studies have shown that patterns of tissue volume loss in TLE follow the brain's structural network embedding, with brain atrophy originating in the mesiotemporal cortices at early stages of the disease, subsequently affecting extratemporal tissues, as well as structurally connected regions in distant neural territories[6,11,12,65]. Our study extends this work by showing that the spatial pattern of functional alterations is also shaped by the underlying structural connectome, reinforcing the fundamental role of global network architecture in shaping cortical abnormalities in TLE. In addition, several studies have demonstrated that brain regions are targeted non-

randomly by disease; those that are highly connected and potentially important for communication tend to be disproportionately affected[66]. Indeed, our epicenter mapping analysis identified key areas implicated in functional disruption, primarily in regions often considered to belong to transmodal association cortex, such as the medial temporal, prefrontal, and parietal regions[6,9]. The association cortex is a core component of the brain's putative rich-club—densely interconnected regions that are thought to facilitate global signal integration and information broadcasting[67]. As network disconnection of the association cortex has been well documented in TLE[8,68], we build upon prior work by showing that medial prefrontal and parietal regions are both affected and likely relate to distributed network-level propagation of functional imbalances. Altogether, our analysis looking beyond regional changes and considering large-scale alterations in brain organization highlights the interconnected nature of TLE pathology, emphasizing that macroscale functional disturbances are strongly shaped by the brain's structural architecture, and motivates efforts to explore the potential of network-level interventions as therapeutic strategies[6,69].

Our multimodal MRI framework enables us to investigate the relationship between functional deviations and structural compromise in the same cohort, with a particular focus on patient-level variability. Notably, we found the most marked deviations in TLE in the ipsilateral temporal lobe, affecting gray matter morphology (CT) and white matter integrity (FA and MD), with varying degrees of severity. Through a multilinear regression model, we revealed a close fit between observed functional deviation maps and maps predicted on the basis of structural deviations, suggesting that structural damage partially accounts for functional alterations in TLE. Unlike earlier studies that primarily explored group-averaged patterns of brain changes[8,45], this study emphasizes coupled interindividual variability in both structural and functional alterations. This individualized approach is highly relevant in TLE, where neuroanatomical changes have shown to vary heavily between patients in both mesiotemporal and neocortical systems[10,47,70]. Notably, additional dominance analysis revealed that MD contributed most to the predictive relationship, underscoring a critical role of white matter integrity in brain function and dysfunction[7,8]. White matter pathways serve as the structural backbone for effective communication between brain regions, facilitating the transmission of neural signals, synchronization of activity, and coordination of functional networks[71,72]. Accordingly, white matter microstructural impairment, as observed in TLE, may disrupt local brain activation or long-range functional connectivity, contributing to the breakdown of large-scale networks such as the default mode and limbic networks, which are commonly implicated in TLE[8]. Our results suggest that cortical functional deviations in TLE are not merely reflective of group-level structural changes across patients, but are closely linked to each patient's unique pattern of white matter damage. This patient-specific perspective highlights the importance of personalized assessments in capturing diverse structural and functional profiles of TLE, as well as in understanding pathophysiological mechanisms driving disease progression in each patient. Here, we provide a framework for future studies to track the evolution of individual-level brain structural and functional changes over time, enabling ongoing disease monitoring. Capturing each patient's unique pattern can therefore deepen our understanding of the structure-function interplay underlying TLE at the individual patient level.

Normative modeling shows great promise for personalized clinical decision-making in TLE. Leveraging supervised machine learning techniques, we provided evidence that individualized maps of functional deviations distinguished TLE patients from those with FCD-related extratemporal epilepsy and identified TLE seizure onset zone laterality with high accuracy. The differentiation between left and right TLE patients was primarily driven by functional deviations in the temporal and parietal cortices ipsilateral to the seizure focus.

Specifically, left TLE exhibited greater deviations in the temporal lobe, which is in alignment with known decreases in functional connectivity of the lateral temporal lobe[73]. It remains to be determined whether these observations overall relate to differences in clinical and cognitive phenotypes, and whether they suggest variable pathophysiological substrates of left versus right TLE[54,74]. The approach developed here may inform and ultimately enhance the presurgical evaluation of pharmacoresistant TLE using non-invasive MRI techniques, which have traditionally primarily focused on structural MRI[75-77]. Future translational research should explore the combined utility of structural and functional MRI data to further improve clinically relevant diagnostics.

In the subgroup of TLE patients who underwent temporal lobe resections, we observed a significant, yet modest association between preoperative functional alterations and postoperative seizure outcome. Specifically, patients with postsurgical seizure recurrence showed greater functional deviations in the temporal lobe (including the hippocampus) contralateral to the seizure focus. That is, more distributed abnormalities, particularly those extending to the contralateral hemisphere, are related to unfavorable surgical outcomes. This observation is consistent with clinical intuition[78,79], and prior studies reporting that aberrant functional connectivity of the contralateral temporal cortex is associated with a higher likelihood of seizure recurrence[80,81]. Persistent epileptogenic activity in non-resected brain regions—including those outside the surgical target—can contribute to surgical failure[9,80]. These findings also relate to the broader concept of "temporal plus" epilepsy[82], which refers to cases in which the epileptogenic zone extends beyond the mesiotemporal lobe to involve additional areas. While this usually relates to a broader involvement of ipsilateral regions, such as the insula and temporoparietal junction, surgical failure can also relate to an involvement of contralateral structures[78,82]. Our findings highlight that contralateral functional anomalies may represent an extended substrate that could contribute to seizure relapse after unilateral surgery. Diagnosis and preoperative evaluation of pharmacoresistant epilepsy is most effective when conducted through a multidisciplinary approach[83], combining clinical, electrophysiological, and neuroimaging data. Our machine learning models, informed by individualized functional deviations, predicted postsurgical seizure outcome with moderate accuracy. While not sufficient for independent clinical decision-making, such biomarkers may offer complementary value when interpreted alongside established tools, including structural MRI, PET, electroclinical assessments, and intracranial EEG in more complex cases. As the field moves toward more personalized, network-based models of epilepsy, integrating functional MRI-derived metrics into multimodal frameworks may enhance outcome prediction and improve surgical candidate stratification. Nonetheless, further work is needed to refine and validate these methods—particularly through prospective studies and harmonized protocols—before they can be reliably translated into clinical practice.

Several limitations of our study should be acknowledged. First, its cross-sectional design precludes inferences about disease progression. While we found widespread structural and functional brain deviations in TLE patients, it remains unknown whether these alterations reflect progressive disease processes or stable sequelae of an initial inciting event. Longitudinal studies can help assess the prognostic values of these MRI markers and are necessary to determine whether TLE patients with more deviations at baseline are more likely to experience faster clinical decline. Such efforts are essential to disentangle true progression from static effects of earlier disease onset or injury, and to dissociate disease effects from typical lifespan effects. Second, several barriers hinder the clinical translation of advanced neuroimaging approaches. These include the lack of standardized MRI acquisition protocols across centers—especially for functional MRI in epilepsy—as well as limited access to image processing pipelines and technical expertise in many clinical settings. Moreover, clinical sites may lack healthy control cohorts scanned with the same parameters as their patients, posing a major challenge for developing normative models and quantifying brain deviations at the individual level. Some of these obstacles may be addressed by adopting open-access software, developing normative models that can generalize across scanners and populations, making benchmark datasets openly available, and fostering collaboration and knowledge exchange between neuroimaging and clinical experts.

## Methods
### Participants
We studied 282 individuals with TLE (133 males; mean ± SD age = 32.3 ± 10.0 years, range: 18–64 years), 45 individuals with extratemporal focal cortical dysplasia (FCD) (23 males; 27.2 ± 7.9 years [18–54 years]), and 298 healthy adults (152 males; 29.3 ± 8.0 years [18–60 years]). Participants were drawn from four independent datasets of three epilepsy centers: (i) Montreal Neurological Institute-Hospital (*MICA-MICs*: TLE/healthy controls/FCD = 57/100/17; *NOEL*: 72/42/28)[35], (ii) Universidad Nacional Autónoma de México (*EpiC*: 29/34)[19]; and (iii) Jinling Hospital (*Nanj*: 124/122)[36]. Epilepsy specialists at each center diagnosed patients according to the seizure and syndrome classifications of the International League Against Epilepsy (ILAE)[84]. All patients underwent comprehensive multidisciplinary evaluation, including clinical history, ictal semiology, continuous video-EEG telemetry (both ictal and interictal EEG), as well as clinical MRI (and PET if available), to localize the seizure focus. No TLE patient had a history of traumatic brain injury, a mass lesion (tumors, malformations of cortical development, vascular malformations), or encephalitis. We excluded patients with comorbid neurological or psychiatric disorders and those with a diagnosis of bilateral TLE. 75.9% of TLE patients (*n* = 214) and 88.9% of FCD patients (*n* = 40) had drug-resistant seizures. In TLE, the mean age at seizure onset was 17.8 ± 10.7 years [0.3–60 years], with a mean disease duration of 14.4 ± 11.0 years [0.8–49 years]. For FCD patients, the age at seizure onset was 11.5 ± 6.9 years [0.3–27 years], and the disease duration was 15.9 ± 10.3 years [0.5–48 years]. At the time of analysis, 99 TLE patients had undergone temporal lobe resection as a treatment for their seizures at the Montreal Neurological Institute-Hospital or Jinling Hospital. With a mean follow-up duration of 44 ± 34 months, 74 patients (75%) were completely seizure-free (Engel I), while 25 patients (25%) continued to experience seizures (Engel II–IV). Among all TLE patients, 68.1% (*n* = 192) had ipsilateral hippocampal sclerosis (HS) (Fig. 1A), confirmed by radiological MRI reports, quantitative analyses of T1-weighted and FLAIR images, and, when available, postoperative histological examination[85–87]; 6.4% (*n* = 18) had ipsilateral hippocampal gliosis; and 25.5% (*n* = 72) were MRI-negative. This study was approved by the Ethics Committees of each participating center (*MICA-MICs* and *NOEL*: Montreal Neurological Institute and Hospital, McGill University; *EpiC*: Institute of Neurobiology, Universidad Nacional Autónoma de Mexico; *Nanj*: Jinling Hospital, Nanjing University School of Medicine). All participants provided written informed consent in accordance with the Declaration of Helsinki.

### MRI processing
Multimodal MRI data, including T1-weighted, diffusion MRI, and rs-fMRI, were acquired with 3.0-Tesla scanners (*MICA-MICs*: Siemens Prisma; *EpiC*: Philips Achieva; *Nanj* and *NOEL*: Siemens Trio) in all individuals (prior to surgery for patients). MRI data of the four datasets were preprocessed uniformly using *micapipe* (version 0.2.3; http://micapipe.readthedocs.io) and mapped to cortical surface points (henceforth, vertices)[88]. We extracted the rs-fMRI BOLD time series, cortical gray matter morphology, and white matter microstructural features for each participant using the same set of 360 cortical parcels (HCP-MMP1.0 atlas)[89]. Full details on MRI data acquisition and pre-processing are provided in the **Supplementary Materials**.

## Resting-state fMRI metrics

Signal variability (SV), a local metric of neural signal fluctuations over time, was defined as the root mean squared successive difference between time points of rs-fMRI time series[90]:

$$SV = \frac{\sqrt{\sum_{i=1}^{T-1}(x_{i+1} - x_i)^2}}{T-1} \quad (1)$$

where $x_i(t)$ is the rs-fMRI signal at the $i$-th time point for a node $x$, with time series length $T$: $x = (x_1, x_2, ..., x_T)$. Higher/lower SV indicates stronger/weaker temporal fluctuations. Although SV has been widely used to index local neural function in both healthy and clinical populations[90–93], no systematic assessment has yet been conducted in epilepsy.

Regional homogeneity (ReHo), as an index of regional brain function, was computed as the temporal coherence of unsmoothed rs-fMRI time series within a given vertex's nearest regional neighbors[94]:

$$ReHo = 12 \frac{\sum_{i=1}^{T} R_i^2 - T(\bar{R})^2}{K^2(T^3 - T)} \quad (2)$$

where $R_i$ is the ranks of time series $x_i(t)$, and $\bar{R}$ is the mean rank across all $K$ neighbors and time points. Higher/lower ReHo represents stronger/weaker local functional homogeneity. ReHo has been widely applied to assess regional subnetwork coherence in both healthy individuals and clinical populations, including individuals with TLE[49,94].

To assess the global connectivity of a given vertex, node strength (NS) was defined as the sum of the weighted functional connectivity strengths, calculated as the Pearson correlation coefficient between rs-fMRI time series of a given brain region and all other regions[95]:

$$NS = \sum_{j=1, j \neq i}^{n} \text{corr}(x(t), y_j(t)) \quad (3)$$

where $y_j(t)$ represents the rs-fMRI time series for a node, excluding node $i$, and $n = 360$ is the total number of nodes. A higher NS indicates greater similarity in intrinsic neural activity between regions, whereas a lower NS reflects greater inter-regional dissimilarity. In both the study of healthy and neurological patient populations[33,96], NS has been extensively used to chart globally interconnected hub regions and their disruption. To further explore the influence of connection distance on NS, we computed short-range and long-range NS for each brain region based on Euclidean distance. The cut-off distance is typically 75 mm[38–40]. Short-range NS was defined as the sum of functional connectivity strengths between a given region $i$ and regions located within 75 mm. Long-range NS was computed as the sum of functional connectivity strengths between region $i$ and all regions located more than 75 mm away.

## Individualized W-score maps

To reduce potential biases and non-biological variability induced by sites/scanners, we used ComBat to harmonize each rs-fMRI metric independently across participants[97]. ComBat uses multivariate linear mixed-effects regression and empirical Bayes to correct for batch effects[98]. It has previously been applied to eliminate site effects in multi-site rs-fMRI studies[99,100]. In this work, the dataset was treated as a batch variable, while preserving variances of interest, including age, sex (males, females), and diagnosis (controls, TLE, extra-TLE). The harmonized metrics, corrected for inter-site variations, were subsequently used to generate W-score maps.

For each metric and each brain region, a W-score was calculated using the healthy control group as a reference[25,37]. Analogous to the Z-score[28] with a mean value of 0 and a standard deviation of 1 relative to healthy controls, the W-score reflects normalized deviation of the observed value from the normative range for healthy adults of the same age and sex. Here, age- and sex-related changes in each rs-fMRI metric were estimated in the healthy control group through a linear regression model ($\beta_0 + \beta_1 \times$ age $+ \beta_2 \times$ sex $+$ residuals) (Fig. 1C), yielding beta maps for intercept value ($\beta_0$), age ($\beta_1$) and sex-related ($\beta_2$) coefficients, as well as individual maps of residuals for each brain region. Patient-specific W-scores were estimated using the formula[25,37]:

$$W\,score = \frac{\text{TLE}_{observed} - \text{TLE}_{predicted}}{\text{SD}(\text{HC}_{observed} - \text{HC}_{predicted})} \quad (4)$$

where, the value predicted for a given patient's age and sex ($\text{TLE}_{predicted}$) was calculated as: $\beta_0 + \beta_1 \times$ age $+ \beta_2 \times$ sex, and SD represented the standard deviation of model residuals in healthy controls. To this end, deviations from normative distributions of SV, ReHo and NS were generated for each brain region and each patient. Positive and negative W-scores indicate deviations above and below the normative ranges, respectively. Brain regions with W-scores exceeding ±1.96 were classified as showing extreme deviations (above and below the 2.5th percentiles)[29–31]. The proportion of extreme deviations (i.e., group-level deviation rate) was computed by counting patients with |W-scores| ≥ 1.96 and dividing by the total patient count in each brain region and was stratified across five lobes (i.e., frontal, parietal, temporal, occipital, and insular). Finally, to aggregate the three rs-fMRI metrics, we built a composite score for each individual, defined as the Mahalanobis distance of the joint distributions[43], and used the map in our multivariate W-score calculations.

## Neighborhood deviation estimates

A group-averaged structural connectivity matrix, derived from an independent sample of 100 healthy adults, was used to define the neighbors of a given brain region[42]. The collective deviation prevalence value $\bar{p}_i$ for neighbors of the $i$-th brain region was estimated as the average weighted prevalence value of all brain regions structurally connected to region $i$ (i.e., regions with no structural connectivity to region $i$ were excluded)[9,101]:

$$\bar{p}_i = \frac{1}{N_i} \sum_{j=1, j \neq i}^{N_i} p_j \cdot C_{ij} \quad (5)$$

where $N_i$ was the number of connected neighbors of region $i$ (i.e., node degree), $p_j$ was the deviation prevalence value observed in the $j$-th neighbor of region $i$, and $C_{ij}$ was the connectivity strength between $i$ and $j$. Under this model, neighbors of region $i$ with a more strongly weighted connection made stronger contributions to estimating region $i$'s neighborhood brain function change. Spearman correlation was used to assess the relationship between node and mean neighbor prevalence values. Statistical significance of correlation coefficients between cortical maps was determined using spatial autocorrelation-preserving spin tests[102]. This procedure was repeated 5000 times, generating a null distribution of brain maps with preserved spatial autocorrelation. The P-value (i.e., $P_{spin}$) was calculated as the fraction of correlation coefficients in null models that exceeded the empirical correlation coefficient.

## Disease epicenters mapping

Disease epicenters were identified by spatially relating each brain region's healthy structural connectivity profile (from the same independent dataset as above[42]) to the group-level functional deviation prevalence map in TLE[6]. This approach was repeated systematically for each brain region, and the significance of spatial similarity was determined using spin permutation tests with 5000 iterations. Brain regions with significant spatial correlations ($P_{spin} < 0.05$) were ranked in descending order according to the epicenter likelihood (i.e., Spearman correlation coefficients), with highly ranked regions representing potential epicenters. As for the hippocampus and subcortical

structures, cortico-subcortical structural connectivity profiles were systematically compared to the group-level functional deviation prevalence map and assessed the significance of correlations using spin permutation tests.

### Associations with structural deviations

For subject-wise neuroanatomical and microstructural profiling, we derived three features—CT, FA and MD—previously reported as abnormal in neuroimaging studies of TLE, using T1-weighted and diffusion MRI (see Supplementary Materials)[8,16,43,45]. Following the same framework applied to rs-fMRI metrics, we generated individualized $W$-score maps for each feature and the group-level prevalence maps of extreme deviations ($|W\text{-score}| \geq 1.96$).

A multilinear regression model was employed to assess the associations between structural and functional deviations across the neocortex (Fig. 4A):

$$y = \beta_0 + \beta_1 \cdot \mathrm{CT} + \beta_2 \cdot \mathrm{FA} + \beta_3 \cdot \mathrm{MD} \qquad (6)$$

where the dependent variable was the group-level functional deviation prevalence map, and the independent variables were structural deviation prevalence maps. The intercept $\beta_0$ and regression coefficients ($\beta_1$, $\beta_2$, $\beta_3$) were optimized to maximize the spatial correlation between the empirical and predicted functional deviation prevalence maps. Model fit was quantified using the adjusted-$R^2$ (coefficient of determination).

Dominance analysis was conducted to determine the relative contributions of structural metrics to the prediction of brain function deviation prevalence[103]. Briefly, dominance analysis estimated the relative importance of each independent variable by constructing all possible combinations of variables and refitting the multilinear regression model for every combination. The contribution of each variable was quantified as the increase in explained variance (i.e., gain in adjusted-$R^2$) when that variable was added to the model. Dominance scores were normalized by total model fit, enabling comparisons across models.

### Clinical utility of functional deviations

Linear support vector machine (SVM) models, implemented in LIBSVM[104], were used to evaluate the utility of individual functional deviation scores in classifying diagnoses, seizure onset zone laterality, and postsurgical outcome. Model training and testing used a nested 5-fold cross-validation with 1000 iterations. Specifically, in each outer iteration, 4 folds were used for training and the remaining fold for testing until all folds had served as training and testing sets. An inner 5-fold cross-validation loop was applied to the training set to optimize hyperparameters ($c$, $g$) through grid search. Once the hyperparameters were determined, models were retrained on the entire training set. Model performance was evaluated on the testing set using the area under the receiver operating characteristic curve (AUC). Statistical significance was determined using 1000 permutation tests with randomly shuffled participant labels, with the $P$-value (i.e., $P_{\mathrm{perm}}$) calculated as the fraction of AUC values in null models that exceeded the empirical AUC.

**TLE versus non-TLE classification.** To assess the utility of functional deviation scores in identifying diagnostic categories, we trained a linear SVM model to distinguish TLE patients ($n = 129$) from extra-temporal FCD patients ($n = 45$) in two datasets (MICs and NOEL) that had both cohorts available. Stratified splitting was employed to ensure consistent TLE/FCD ratios across all data splits.

**Seizure focus zones lateralization.** To evaluate the effectiveness of functional deviation scores in identifying seizure onset zone laterality, we trained a linear SVM model to classify left TLE patients ($n = 147$) versus right TLE patients ($n = 135$) using stratified splitting to preserve the left/right TLE ratio across all splits.

**Postsurgical seizure outcome prediction.** Postsurgical seizure outcome data were available for 99 TLE patients from 3 datasets (MICs, NOEL, and Nanj) who underwent temporal lobe resections: 74 (75%) were seizure-free (Engel class I), 25 (25%) were not seizure-fee (Engel class II–IV). Postsurgical MRI scans were available in 35 patients, enabling the segmentation of patient-specific surgical cavity masks from pre- and post-operative T1w MRIs. We first examined the spatial correspondence between the pattern of functional deviations and the pattern of between-patient overlap in surgical cavities, defined as the proportion of patients with resection in each region (out of 35 patients); next, we compared the proportion of extreme deviations (i.e., number of deviations divided by the total number of regions in each lobe) between the ipsilateral and contralateral hemispheres in each lobe for each TLE subgroup (seizure-free and non-seizure-free) using paired $t$-tests ($P_{\mathrm{FDR}} < 0.05$); also, we trained a linear SVM model to classify seizure-free versus non-seizure-free TLE patients based on individual functional deviation scores using stratified splitting.

**Associations with clinical characteristics.** Associations between the number of extreme deviations and disease course parameters (age at seizure onset, disease duration, and number of antiseizure medications) were evaluated in TLE patients using bivariate correlations. In addition, we split TLE patients according to the presence or absence of focal to bilateral tonic–clonic seizures and compared the number of extreme deviations between the two subgroups using two-sample $t$-tests.

### Sensitivity analyses

**Left and right TLE.** As the laterality of seizure focus may differentially affect the distribution of brain functional anomalies, we repeated calculating the group-level prevalence of extreme functional deviations in left ($n = 147$) and right ($n = 135$) TLE subgroups independently.

**Head motion and global mean signal.** To assess the effects of head motion and global signal regression (GSR) on rs-fMRI data, we repeated the $W$-score analyses while additionally controlling for individual head motion (i.e., mean framewise displacement) during rs-fMRI scans, or using GSR-preprocessed data.

### Reporting summary

Further information on research design is available in the Nature Portfolio Reporting Summary linked to this article.

## Data availability

Raw data from the MICA-MICs dataset are available via the Canadian Open Neuroscience Platform (CONP: https://portal.conp.ca/) and the Open Science Framework (OSF: https://osf.io/j532r/). Raw data from the EpiC dataset are available on OpenNeuro (data set ds004469, https://openneuro.org/datasets/ds004469/versions/1.1.4). Access to raw data from the Nanj and NOEL datasets can be granted upon reasonable request, subject to institutional review and the signing of a data use agreement outlining the terms of access, use, storage and authorship. Source data are provided with this paper.

## Code availability

MRI preprocessing was conducted using micapipe (v0.2.3; http://micapipe.readthedocs.io). Code for spin permutation testing is available at https://github.com/frantisekvasa/rotate_parcellation. Code for dominance analysis is available at https://netneurolab.github.io/netneurotools. The LIBSVM toolbox used for supervised machine learning is available at https://github.com/cjlin1/libsvm (v3.31). The visualization of brain mapping images was conducted using the ENIGMA Toolbox (v2.0.1; https://enigma-toolbox.readthedocs.io). Costume scripts used for analysis are available at https://github.com/MICA-MNI/TLE_rsfmri-deviations.

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

## Acknowledgments

K.X. is funded by the China Scholarship Council, Healthy Brains and Healthy Lives (HBHL) Doctoral Fellowship, and the Savoy Foundation. E.S. and J.C. are supported by the Canadian Institutes of Health Research (CIHR) Vanier Canada Graduate Scholarships. A.N. and J.R. are funded by the CIHR. R.R.C. is funded by the Fonds de Recherche du Québec—Santé (FRQ-S). S.L. is funded by the Center de Recherche du CHUS. Lo.C. acknowledges support from a Berkeley Fellowship. Lu.C. is funded by the Consejo Nacional de Ciencia y Tecnología (CONACYT) (181508, 1782, FC218-2023) and Dirección General de Asuntos del Personal Académico (DGAPA) - UNAM (IB201712, IG200117, IN204720, IN213423). B.C.B. acknowledges research support from the National Science and Engineering Research Council of Canada (NSERC RGPIN-2025-05932), CIHR (FDN-154298, PJT-174995, PJT-191853), SickKids Foundation (NI17-039), Helmholtz International BigBrain Analytics and Learning Laboratory (HIBALL), HBHL, Brain Canada Foundation, FRQS, Tier-2 Canada Research Chairs Program, and The Center for Excellence in Epilepsy at the Neuro (CEEN).

## Author contributions

Study Concept/Design: K.X., A.B., N.B., and B.C.B. Data Acquisition, Analysis, Interpretation: K.X., E.S., A.N., J.C., T.A., J.R., Y.Z., R.R.C., A.D., B.C., F.F., S.L., R.P., B.F., D.V.S., Z.Z., Lu.C., A.B., N.B., and B.C.B. Writing – Original Draft: K.X. and B.C.B. Writing – Review & Editing: K.X., E.S., A.N., J.C., T.A., J.R., Y.Z., R.R.C., A.D., B.C., F.F., Alex.B., S.A., S.L., Lo.C., R.P., A.G.W., C.G., B.F., D.V.S., Z.Z., Lu.C., A.B., N.B., and B.C.B. Resources: Z.Z., Lu.C., A.B., N.B., and B.C.B. Supervision: A.B., N.B., and B.C.B.

## Competing interests

The authors declare no competing interests.

## Additional information

¹McConnell Brain Imaging Centre, Montreal Neurological Institute and Hospital, McGill University, Montreal, QC, Canada. ²Sherbrooke Laboratory for Integrative Connectomics, Centre de Recherche du CHUS, Université de Sherbrooke, Sherbrooke, QC, Canada. ³Department of Neurology, Inselspital, Sleep-Wake-Epilepsy-Center, Bern University Hospital, University of Bern, Bern, Switzerland. ⁴Montreal Neurological Institute and Hospital, McGill University, Montreal, QC, Canada. ⁵Division of Neurosurgery, Department of Surgery, Sainte-Justine University Hospital Centre, Montreal, QC, Canada. ⁶Multimodal Functional Imaging Lab, Department of Physics and Concordia School of Health, Concordia University, Montreal, QC, Canada. ⁷Multimodal Functional Imaging Lab, Department of Biomedical Engineering, McGill University, Montreal, QC, Canada. ⁸Department of Neurology, Department of Biomedical Engineering, Duke University, Durham, NC, USA. ⁹British Columbia Children's Hospital, University of British Columbia, Vancouver, BC, Canada. ¹⁰Department of Medical Imaging, Jinling Hospital, Nanjing University School of Medicine, Nanjing, China. ¹¹Institute of Neurobiology, Universidad Nacional Autónoma de Mexico, Queretaro, Mexico. ¹²These authors jointly supervised this work: Andrea Bernasconi, Neda Bernasconi, Boris C. Bernhardt. ✉e-mail: ke.xie@mail.mcgill.ca; boris.bernhardt@mcgill.ca

