## [Transparent Peer review file · Nature Communications]

Personalized Biomarkers of Multiscale Functional Alterations in Temporal Lobe Epilepsy

Corresponding Author: Dr Boris Bernhardt

Version 0:

Reviewer comments:

Reviewer #1

(Remarks to the Author)

The authors studied a cohort of patients with drug resistant temporal lobe epilepsy and compared their resting state fMRI data to a cohort of health controls as well as a small cohort of patients with extra-temporal lobe epilepsy. They used resting state BOLD based measures of FC across different scales (i.e., local, regional, global FC) and determined the degree of deviation in these measures from age and sex matched norms derived from the control cohort, so called spatial normative modeling. They found deviations in TLE across all measures. They additionally found that the FC deviations were in part constrained by structural connectivity using normative connectomes and patient-specific measures of superficial WM. FC deviations performed reasonably well (in SVM models) in differentiating TLE from extra-TLE, epilepsy lateralization, and post-surgical outcome. I found the motivation and overall analysis sound although some elaboration of the motivation for certain analytic decisions could be better explained so the manuscript is more accessible to the reader who is not familiar with the group's work. I found the discussions section, interesting; however, the authors imply in several areas that normative approaches to FC in TLE are ready for clinical translation, which needs to be tempered. While this could be translatable if validated, there is no discussion of barriers to clinical translation. Below I outline some additional considerations:

I had trouble understanding what was done in Section 2.3 Network Constraints. In previous work by the group using ENIGMA morphometry data and normative connectomes, they mapped group level regional cortical thickness in TLE patients to that regions normative connectivity inferred from HCP data. Was the same approach followed here? The authors should elaborate a bit more what was done and the motivation for doing so.

In Section 2.4, Is the FA and MD extracted from the superficial WM surface? The reader needs to scroll all the way to Methods to find this information. What was the motivation for using this measure? I know the authors have used this measure in prior work, but it would be helpful to the general reader to explain the motivation a little better.

There is no Limitations section. What are the barriers to clinical translation? Most centers do not have healthy control cohorts to norm their patients to.

How did you handle the harmonization given that one site did not have extra-TLE patients? Did you use a categorical variable for epilepsy vs control in Combat?

Which kernel was used for SVM models?

(Remarks on code availability)

Reviewer #2

(Remarks to the Author)

This study evaluates multiscale functional alterations using MRI-based data, to establish personalized biomarkers for patients with temporal lobe epilepsy. Overall, this study addresses an important issue in clinical epilepsy care, as localization of temporal lobe epileptic seizures is critical for important treatments, such as resective epilepsy surgery. Past studies have demonstrated group level trends using MRI-based algorithms for seizure localization, but the heterogeneity of changes within individual patients has limited the positive predictive value of these changes in individualized patient care. The authors bring an impressive track record of developing image processing algorithms for structural and functional MRI post-image acquisition processing in epilepsy. They used many of their past algorithms to combine structural and functional MRI-based analyses to localize abnormalities on an individual patient level.

Overall, the imaging acquisition and post-acquisition analyses are sound. Specific comments below:

- 1) The authors collect data from four different centers, each of course with a different MRI scanner. While the authors mention the important issue of harmonization of images between centers, they do not mention this important issue in the discussion. A brief mention of this issue in the discussion could clarify the importance of harmonization of images.
- 2) Temporal lobe epilepsy with, and without associated hippocampal sclerosis (HS) represent 2 clinically important categories of TLE. Many past functional and structural imaging studies show significant differences between these two temporal lobe epilepsy categories. While the authors document subjects with HS, most of their analyses involve the MTS and non-HS TLE cohort overall. There is mention of TLE with HS in the discussion on line 310 in relationship to changes in the insula. However, a further discussion of the implications of combining the MTS and non-MTS cases, as well as implications for the results could help the reader understand the important pathophysiological and clinical differences in these TLE subtypes.
- 3) Degree of certainty of diagnosis of temporal lobe epilepsy varies among cohorts. Typically, the gold standard for verification of TLE involves either intracranial EEG studies, or response to epilepsy surgery. While there is report of a subset of subjects who underwent epilepsy surgery, other details of specific EEG studies to confirm diagnosis is not presented. Presumably, a good number of patients were localized with scalp EEG studies. A list of EEG verification technique, including whether localization was based on ictal and interictal data, as well as scalp versus intracranial EEG, would better define the cohort. This is important because interictal and scalp EEG studies can incorrectly localize epileptic seizures.
- 4) On line 322 in the discussion, the authors acknowledge the cross-sectional nature of their study. They then proceed to discuss possible progressive changes in epileptic seizures. The overwhelming majority of past imaging studies in clinical epilepsy unfortunately had a cross-sectional, rather than prospective design. While large network involvement in epilepsy subjects is well demonstrated in the study, it does not specifically address progression over time. Additionally, because epilepsy often begins with an inciting event, it remains possible that a more severe inciting event results in longer-lasting seizures, rather than progression of changes over time. The authors should emphasize that progression of changes in functional and structural MRI remains understudied, and not definitively proven.
- 5) The concept of personalizing biomarkers in temporal lobe epilepsy is extremely important. The authors correctly state a modest predictive value of preoperative functional abnormalities on the individual level for prediction of seizure freedom after epilepsy surgery. Therefore, their current techniques for individualized biomarkers can only be used as a supplemental confirmatory test, interpreted in the context of other highly predictive tests, such as intracranial EEG.
- 6) In line 393, the authors mention “temporal plus” epilepsy in relationship to failure of epilepsy surgery. While definitions of “temporal plus” epilepsy can implicate contralateral foci causing seizures, they also include ipsilateral extratemporal foci as a cause for failure of the surgery. A stated definition of “temporal plus” would be helpful.

(Remarks on code availability)

Reviewer #3

(Remarks to the Author)

Thank you for inviting me to review this paper by Xie et al. This paper has several positive elements, including a novel functional-structural brain and normative framework in one of the largest temporal lobe epilepsy cohorts to date. The machine learning model is also constructed well in a nested approach. Despite these strengths of this work, I have several concerns (outlined below) that I hope the authors will find helpful:

The title suggests a personalised biomarker of temporal lobe epilepsy in a cortical parcellation mask: Despite the scientific advance in this work, I am not sure the ‘personalised biomarker’ tag in the title is warranted based on the result. One of the reasons for this is that large amounts of the subcortical brain is not being analysed in this approach with the cortical Glasser 360 parcellation mask (<https://www.nature.com/articles/nature18933>). It would be great to hear why the subcortical brain was excluded from this work, particularly in a neurological condition where many people have a seizure focus from the mesial temporal lobes. Several subcortical masks are now available that may be convolved with the Glasser mask for full brain coverage: e.g., <https://elifesciences.org/articles/59430> & <https://pubmed.ncbi.nlm.nih.gov/32989295/> & <https://www.researchsquare.com/article/rs-4203104/v1>. Secondly, it is unlikely that machine learning results with AUC of

0.63-0.76 are strong enough to progress to individual biomarkers without further development.

Defining local, regional and global connectivity: Determining the spatial extent of correlations is challenging, especially when averaged across a larger brain area in a parcellation framework. Given its cubical constraints, I see the rationale for calling ReHo a regional measure, but NS is challenging. It would be good to see an Euclidean distance measure for NS to confirm that we are looking at global changes in the brain rather than reflecting correlations within spatially proximate nodes. Also, signal variability and power spectral density are intrinsically related, so it would be great to hear how the SV algorithm correlates with ALFF/fALFF (<https://pmc.ncbi.nlm.nih.gov/articles/PMC3902859/>)

Multisite data without scanner harmonization? I found no information about applying between-scanner harmonization correction in these cohorts. Large amount of MRI research suggests that systematic scanner effects are an issue, but can be (satisfactory) corrected with approaches such as ComBat (<https://pmc.ncbi.nlm.nih.gov/articles/PMC6179920/>). Recent research indicates that large fMRI variability between scanners for the same subjects (https://direct.mit.edu/imag/article/doi/10.1162/imag_a_00042/118214/A-resource-for-development-and-comparison-of), so this is likely to be an issue especially for the machine learning analysis as two sites have controls and four sites have epilepsy subjects. In other words, there is an imbalance of groups between scanners. Smoothing, filtering and regressors:

For clarification:

- What filtering band was used for the fMRI connectivity data?
- What was the rationale for using a large smoothing parameter extent (10mm) in this cohort?
- Finally, how many parameters were included in the motion correction (on page 3 of supplementary materials)?

(Remarks on code availability)

Version 1:

Reviewer comments:

Reviewer #1

(Remarks to the Author)

No additional comments. The authors have satisfactorily addressed all my comments.

(Remarks on code availability)

N/A

Reviewer #2

(Remarks to the Author)

The authors have comprehensively addressed my questions about the manuscript. I have no further questions or concerns.

(Remarks on code availability)

Reviewer #3

(Remarks to the Author)

The authors have addressed my previous concerns. This is great work with impact in the field.

(Remarks on code availability)

It would be good to have more data accessible, as it is not possible to run any of the machine learning functions with only two control datasets. The code overall looks ok.

RESPONSE TO REVIEWERS (NCOMMS-25-33479)

We sincerely thank the Editors for the opportunity to submit a revised manuscript, and the Reviewers for their thoughtful evaluations. We greatly appreciate their constructive comments, which have been instrumental in improving the quality of our work. We have addressed all the Reviewers' suggestions in a point-by-point manner and have highlighted the corresponding changes in the revised manuscript in red. All revised figures are included at the end of this document.

Reviewer #1 (Remarks to the Author):

The authors studied a cohort of patients with drug resistant temporal lobe epilepsy and compared their resting state fMRI data to a cohort of health controls as well as a small cohort of patients with extra-temporal lobe epilepsy. They used resting state BOLD based measures of FC across different scales (i.e., local, regional, global FC) and determined the degree of deviation in these measures from age and sex matched norms derived from the control cohort, so called spatial normative modeling. They found deviations in TLE across all measures. They additionally found that the FC deviations were in part constrained by structural connectivity using normative connectomes and patient-specific measures of superficial WM. FC deviations performed reasonably well (in SVM models) in differentiating TLE from extra-TLE, epilepsy lateralization, and post-surgical outcome. I found the motivation and overall analysis sound although some elaboration of the motivation for certain analytic decisions could be better explained so the manuscript is more accessible to the reader who is not familiar with the group's work. I found the discussions section, interesting; however, the authors imply in several areas that normative approaches to FC in TLE are ready for clinical translation, which needs to be tempered. While this could be translatable if validated, there is no discussion of barriers to clinical translation. Below I outline some additional considerations:

We thank the Reviewer for their thoughtful and constructive feedback. We agree that while normative modeling approaches hold translational potential, more work is needed before they are ready for clinical implementation. We revised the **Discussion** to clarify the scope and limitations of our findings. Specifically, we tempered statements related to clinical applicability and explicitly acknowledge barriers to translation, including the need for prospective validation, harmonization across imaging protocols and centers, integration with clinical workflows, and usability testing (**p. 14-15**).

“Several limitations of our study should be acknowledged. First, its cross-sectional design precludes inferences about disease progression. ... Second, several barriers hinder the clinical translation of advanced neuroimaging approaches. ... Moreover, clinical sites may lack healthy control cohorts scanned with the same parameters as their patients, posing a major challenge for developing normative models and quantifying brain deviations at the individual level. ...”

We also expanded the **Results** and **Methods** to provide more background on certain analytic choices to improve accessibility for readers less familiar with normative modeling approaches (**p. 8-9**).

“2.3 Network Constraints on Regional Functional Deviations.

Since brain regions connected to others with high local vulnerability have great disease exposure, we explored the extent to which the spatial pattern of functional changes in TLE is reflected by the white matter connections. ...”

“2.4 Associations with Structural Deviations.

Given that the superficial white matter (SWM) beneath the cortical interface harbors terminations of both short-range U-fibers and long-range tracts,⁴⁴ it serves as an ideal substrate for integrative studies of cortical function and microstructure. ...”

1) I had trouble understanding what was done in Section 2.3 Network Constraints. In previous work by the group using ENIGMA morphometry data and normative connectomes, they

mapped group level regional cortical thickness in TLE patients to that regions normative connectivity inferred from HCP data. Was the same approach followed here? The authors should elaborate a bit more what was done and the motivation for doing so.

In this study, we evaluated whether TLE-related functional deviations are shaped by the architecture of the white matter connectome using two complementary analyses. The first is a new network-based prediction analysis for TLE that quantifies how well a region’s alteration can be inferred from alterations in its structurally connected neighbours. The second is an epicenter-mapping analysis following procedures from our prior ENIGMA-Epilepsy morphological work (Larivière et al., 2020, *Science Advances*). To aid interpretation, we have expanded the **Results** to clarify both approaches (**p. 8**):

“Since brain regions connected to others with high local vulnerability have great disease exposure, we explored the extent to which the spatial pattern of functional changes in TLE is reflected by the white matter connections. Specifically, we tested whether the structural connectivity profile of a given area i could predict the functional deviation score of areas structurally connected to it.^{9,41} For each brain region, we calculated the mean deviation prevalence of its structurally connected neighbors, weighted by white matter connectivity strength estimated through diffusion MRI streamline tractography (**Figure 3A**). To ensure that connectivity estimates reflected typical connectomes, a group-level structural connectivity matrix was generated across 100 unrelated healthy young adults from the Human Connectome Project (HCP).⁴² Statistical significance was assessed with spin permutation tests to account for spatial autocorrelation. We observed a positive correlation between regional deviation scores and the average deviation of their structurally connected neighbors ($\rho = 0.27$, $P_{\text{spin}} < 0.001$), suggesting that the spatial patterning of TLE-related functional alterations is constrained by inter-regional structural connectivity.

Moreover, we identified disease epicenters as brain regions whose structural connectivity profiles closely resembled the spatial pattern of TLE-related functional alterations. To this end, we computed spatial correlations between each region’s structural connectivity profile and the functional deviation map in TLE. Accordingly, the higher the correlation, the more likely the region represented a disease epicenter (**Figure 3B**). Statistical significance was assessed using non-parametric permutation tests (5,000 iterations), and regions with significant correlations ($P_{\text{spin}} < 0.05$) were identified as potential epicenters.⁶ Top-ranked neocortical epicenters included the ipsilateral mesiotemporal cortex and bilateral medial prefrontal and parietal cortices (**Figure 3C**). Among subcortical and mesiotemporal structures, the bilateral hippocampi, pallidum, and nucleus accumbens, along with the ipsilateral caudate, emerged as dominant epicenters (**Figure 3D**).”

2) In Section 2.4, Is the FA and MD extracted from the superficial WM surface? The reader needs to scroll all the way to Methods to find this information. What was the motivation for using this measure? I know the authors have used this measure in prior work, but it would be helpful to the general reader to explain the motivation a little better.

As suggested, we clarified that FA and MD were extracted from the superficial white matter surface, and provided further rationale for these measures (**p. 9**):

“We next examined the association between cortical functional and structural alterations in TLE using MRI markers of whole-brain structural compromise commonly employed in prior studies of this condition.^{8,16,43} Cortical thickness (CT) was estimated as the Euclidean distance between corresponding vertices on the pial surface and the grey-white boundary, derived from processed T1-weighted MRI. Given that the superficial white matter (SWM) beneath the cortical interface harbors terminations of both short-range U-fibers and long-range tracts,⁴⁴ it serves as an ideal substrate for integrative studies of cortical function and microstructure. We extracted SWM features—fractional anisotropy (FA) and mean diffusivity (MD)—from diffusion MRI as surrogates of fiber architecture and tissue microstructure. Analogy to brain functional metrics, individualized structural deviations (*i.e.*, CT, FA, and MD) were quantified by positioning each TLE patient’s data onto normative percentile charts derived from healthy controls. Across the cortex, structural deviations were most pronounced in the temporal lobe (mean CT/FA/MD = 7.5%/14.7%/16.6%), followed by the insular cortex (8.2%/9.6%/10.6%) and, to a lesser extent, the parietal lobe (7.8%/7.6%/10.1%; **Figure**

4A). To assess the relationship between structural and functional abnormalities, we employed a multilinear regression model to predict the cortex-wide prevalence of functional deviations based on the prevalence of structural deviations. We observed a positive correlation between empirical and predicted functional deviation patterns, above and beyond the effect of spatial autocorrelation ($\rho = 0.27$, $P_{\text{spin}} < 0.001$; **Figure 4B**). Dominance analysis further revealed that MD contributed most substantially to the model fit (56%), with modest contributions from CT (24%) and FA (20%) (**Figure 4C**).”

3) There is no Limitations section. What are the barriers to clinical translation? Most centers do not have healthy control cohorts to norm their patients to.

As recommended, we included a **Limitation** section in the revised **Discussion** to acknowledge methodological constraints and barriers to clinical translation (p. 14-15):

“Several limitations of our study should be acknowledged. First, its cross-sectional design precludes inferences about disease progression. While we found widespread structural and functional brain deviations in TLE patients, it remains unknown whether these alterations reflect progressive disease processes or stable sequelae of an initial inciting event. Longitudinal studies can help assess the prognostic values of these MRI markers and are necessary to determine whether TLE patients with more deviations at baseline are more likely to experience faster clinical decline. Such efforts are essential to disentangle true progression from static effects of earlier disease onset or injury, and to dissociate disease effects from typical lifespan effects. Second, several barriers hinder the clinical translation of advanced neuroimaging approaches. These include the lack of standardized MRI acquisition protocols across centers—especially for functional MRI in epilepsy—as well as limited access to image processing pipelines and technical expertise in many clinical settings. Moreover, clinical sites may lack healthy control cohorts scanned with the same parameters as their patients, posing a major challenge for developing normative models and quantifying brain deviations at the individual level. Some of these obstacles may be addressed by adopting open-access software, developing normative models that can generalize across scanners and populations, making benchmark datasets openly available, and fostering collaboration and knowledge exchange between neuroimaging and clinical experts.”

4) How did you handle the harmonization given that one site did not have extra-TLE patients? Did you use a categorical variable for epilepsy vs control in Combat?

To correct for variability related to scanners/sites, we applied ComBat harmonization to each rs-fMRI metric independently, using dataset as the batch variable. In the ComBat model, we included age, sex (male/female), and diagnosis group (controls, TLE, extra-TLE) as biological covariates to preserve relevant inter-individual and group-level variability. This approach allows harmonization across datasets while maintaining variance attributable to key clinical and demographic factors, even when some diagnostic subgroups (*e.g.*, extra-TLE patients) were not represented in all sites. We clarified this point in the revised **Methods** (p. 16):

“To reduce potential biases and non-biological variability induced by sites/scanners, we used ComBat to harmonize each rs-fMRI metric independently across participants.⁹⁷ ComBat uses multivariate linear mixed-effects regression and empirical Bayes to correct for batch effects.⁹⁸ It has previously been applied to eliminate site effects in multi-site rs-fMRI studies.^{99,100} In this work, dataset was treated as a batch variable, while preserving variances of interest, including age, sex (males, females), and diagnosis (controls, TLE, extra-TLE). The harmonized metrics, corrected for inter-site variations, were subsequently used to generate W -score maps.”

5) Which kernel was used for SVM models?

In our study, we used a **linear kernel** for all SVM models. This choice was motivated by its simplicity, strong interpretability, and suitability for datasets with a limited sample size relative to the number of features. Linear SVMs also reduces the risk of overfitting. We clarified this point in the revised **Methods** (p. 18):

“Linear support vector machine (SVM) models, implemented in LIBSVM,¹⁰⁴ were used to evaluate the utility of

individual functional deviation scores in classifying diagnoses, seizure onset zone laterality, and postsurgical outcome.”

Reviewer #2 (Remarks to the Author):

This study evaluates multiscale functional alterations using MRI-based data, to establish personalized biomarkers for patients with temporal lobe epilepsy. Overall, this study addresses an important issue in clinical epilepsy care, as localization of temporal lobe epileptic seizures is critical for important treatments, such as resective epilepsy surgery. Past studies have demonstrated group level trends using MRI-based algorithms for seizure localization, but the heterogeneity of changes within individual patients has limited the positive predictive value of these changes in individualized patient care. The authors bring an impressive track record of developing image processing algorithms for structural and functional MRI post-image acquisition processing in epilepsy. They used many of their past algorithms to combine structural and functional MRI-based analyses to localize abnormalities on an individual patient level.

We thank the Reviewer for the positive evaluation and for the constructive feedback. We answered each of the thoughtful comments one-by-one below.

1) The authors collect data from four different centers, each of course with a different MRI scanner. While the authors mention the important issue of harmonization of images between centers, they do not mention this important issue in the discussion. A brief mention of this issue in the discussion could clarify the importance of harmonization of images.

As the Reviewer rightfully points out, multi-site studies face potential confounds related to scanner/site-specific variability. In our study, we implemented ComBat harmonization to correct for site effects while preserving biologically meaningful between-subject variance. In response, we included a brief discussion of the importance of image harmonization in the revised **Discussion (p. 12)**:

“Encouragingly, we found that patterns of functional brain deviations were highly consistent across sites and robust to the inclusion of global signal regression as an additional denoising step. A key strength of our study lies in the harmonization of preprocessing and analytic workflows across centers, alongside statistical adjustment for site effects. This step enhances the reliability and generalizability of our multi-site findings by minimizing nonbiological variability arising from scanner and acquisition protocol differences.⁵⁸”

2) Temporal lobe epilepsy with, and without associated hippocampal sclerosis (HS) represent 2 clinically important categories of TLE. Many past functional and structural imaging studies show significant differences between these two temporal lobe epilepsy categories. While the authors document subjects with HS, most of their analyses involve the MTS and non-HS TLE cohort overall. There is mention of TLE with HS in the discussion on line 310 in relationship to changes in the insula. However, a further discussion of the implications of combining the MTS and non-MTS cases, as well as implications for the results could help the reader understand the important pathophysiological and clinical differences in these TLE subtypes.

While our main analyses focused on the full cohort to provide a broad characterization of functional changes across the spectrum of TLE, we also conducted a subgroup comparison between HS and non-HS patients (**Figure S5**). This analysis indicates stronger involvement of limbic and subcortical structures in HS-TLE compared to non-HS TLE. We clarified these findings more explicitly in the main text and expand the **Discussion** to outline the potential implications of combining HS and non-HS cases (**p. 12**):

“While the subgroup of patients with HS-TLE had an overall pattern of functional alterations that was broadly consistent with the full cohort, they exhibited increased involvement of limbic and subcortical structures in the ipsilateral

hemisphere (particularly the insula, thalamus and pallidum). These results are consistent with previous MRI-based studies that have reported widespread grey and white matter abnormalities,^{4,52-54} as well as a more pronounced alteration in hippocampal functional connectivity,⁵⁵ network controllability,⁵⁶ and excitation/inhibition imbalance in HS-TLE.⁹ Additionally, they align with *post-mortem* work showing that extrahippocampal pathology in HS-TLE often extends beyond mesiotemporal regions into temporopolar, frontopolar and orbitofrontal cortices.⁵⁷ Altogether, these results support the view that HS-TLE is associated with a more severe disruption of large-scale brain networks. While our main analyses focused on the full cohort to provide a broad characterization of functional alterations across the spectrum of TLE, our subgroup results underscore the modulation of network alterations by the degree of pathological alterations in the mesiotemporal disease epicenter.”

3) Degree of certainty of diagnosis of temporal lobe epilepsy varies among cohorts. Typically, the gold standard for verification of TLE involves either intracranial EEG studies, or response to epilepsy surgery. While there is report of a subset of subjects who underwent epilepsy surgery, other details of specific EEG studies to confirm diagnosis is not presented. Presumably, a good number of patients were localized with scalp EEG studies. A list of EEG verification technique, including whether localization was based on ictal and interictal data, as well as scalp versus intracranial EEG, would better define the cohort. This is important because interictal and scalp EEG studies can incorrectly localize epileptic seizures.

We thank the Reviewer for raising this important point. While not all patients in our study underwent invasive monitoring or resective surgery, all cases were evaluated by experienced epilepsy specialists at each center using a comprehensive multidisciplinary approach, following the seizure and syndrome classification of the ILAE. While intracranial EEG was available only in a subset of patients, TLE diagnosis and seizure focus localization was based on converging evidence from clinical history, semiology, continuous video-EEG telemetry, and clinical imaging. We clarified this point in the revised **Methods (p. 15)**:

“Epilepsy specialists at each center diagnosed patients according to the seizure and syndrome classifications of the International League Against Epilepsy (ILAE).⁸⁴ All patients underwent comprehensive multidisciplinary evaluation, including clinical history, ictal semiology, continuous video-EEG telemetry (both ictal and interictal EEG), as well as clinical MRI (and PET if available), to localize the seizure focus.”

4) On line 322 in the discussion, the authors acknowledge the cross-sectional nature of their study. They then proceed to discuss possible progressive changes in epileptic seizures. The overwhelming majority of past imaging studies in clinical epilepsy unfortunately had a cross-sectional, rather than prospective design. While large network involvement in epilepsy subjects is well demonstrated in the study, it does not specifically address progression over time. Additionally, because epilepsy often begins with an inciting event, it remains possible that a more severe inciting event results in longer-lasting seizures, rather than progression of changes over time. The authors should emphasize that progression of changes in functional and structural MRI remains understudied, and not definitively proven.

We agree that most neuroimaging studies in epilepsy have employed a cross-sectional design, which limits the ability to draw conclusions about progressive changes over time. In response, we revised the **Discussion** to more explicitly acknowledge this limitation and to emphasize the need for future longitudinal investigations (**p. 14**):

“Several limitations of our study should be acknowledged. First, its cross-sectional design precludes inferences about disease progression. While we found widespread structural and functional brain deviations in TLE patients, it remains unknown whether these alterations reflect progressive disease processes or stable sequelae of an initial inciting event. Longitudinal studies can help assess the prognostic values of these MRI markers and are necessary to determine whether TLE patients with more deviations at baseline are more likely to experience faster clinical decline. Such

efforts are essential to disentangle true progression from static effects of earlier disease onset or injury, and to dissociate disease effects from typical lifespan effects. Second, several barriers hinder the clinical translation of advanced neuroimaging approaches. These include the lack of standardized MRI acquisition protocols across centers—especially for functional MRI in epilepsy—as well as limited access to image processing pipelines and technical expertise in many clinical settings. Moreover, clinical sites may lack healthy control cohorts scanned with the same parameters as their patients, posing a major challenge for developing normative models and quantifying brain deviations at the individual level. Some of these obstacles may be addressed by adopting open-access software, developing normative models that can generalize across scanners and populations, making benchmark datasets openly available, and fostering collaboration and knowledge exchange between neuroimaging and clinical experts.”

5) The concept of personalizing biomarkers in temporal lobe epilepsy is extremely important. The authors correctly state a modest predictive value of preoperative functional abnormalities on the individual level for prediction of seizure freedom after epilepsy surgery. Therefore, their current techniques for individualized biomarkers can only be used as a supplemental confirmatory test, interpreted in the context of other highly predictive tests, such as intracranial EEG.

We agree that, given the modest predictive value of our models, functional MRI-derived biomarkers should currently be considered as supplemental tools rather than stand-alone diagnostic measures. In response, we revised the **Discussion** to explicitly acknowledge this limitation and clarify that our approach is intended to complement, rather than replace, established modalities such as structural MRI and electroclinical assessments. We also highlight the need for further validation and integration of such biomarkers within multimodal, prospective clinical workflows (**p. 14**):

“Diagnosis and preoperative evaluation of pharmacoresistant epilepsy is most effective when conducted through a multidisciplinary approach,⁸³ combining clinical, electrophysiological, and neuroimaging data. Our machine learning models, informed by individualized functional deviations, predicted postsurgical seizure outcome with moderate accuracy. While not sufficient for independent clinical decision-making, such biomarkers may offer complementary value when interpreted alongside established tools, including structural MRI, PET, electroclinical assessments, and intracranial EEG in more complex cases. As the field moves toward more personalized, network-based models of epilepsy, integrating functional MRI-derived metrics into multimodal frameworks may enhance outcome prediction and improve surgical candidate stratification. Nonetheless, further work is needed to refine and validate these methods—particularly through prospective studies and harmonized protocols—before they can be reliably translated into clinical practice.”

6) In line 393, the authors mention “temporal plus” epilepsy in relationship to failure of epilepsy surgery. While definitions of “temporal plus” epilepsy can implicate contralateral foci causing seizures, they also include ipsilateral extratemporal foci as a cause for failure of the surgery. A stated definition of “temporal plus” would be helpful.

As suggested, we clarified the definition of “temporal plus” epilepsy in the revised **Discussion**. We better contextualized our findings within this broader framework of extended epileptogenic networks and their potential contribution to surgical failure (**p. 14**):

“Persistent epileptogenic activity in non-resected brain regions—including those outside the surgical target—can contribute to surgical failure.^{9,80} These findings also relate to the broader concept of “temporal plus” epilepsy,⁸² which refers to cases in which the epileptogenic zone extends beyond the mesiotemporal lobe to involve additional areas. While this usually relates to a broader involvement of ipsilateral regions, such as the insula and temporoparietal junction, surgical failure can also relate to an involvement of contralateral structures.^{78,82} Our findings highlight that contralateral functional anomalies may represent an extended substrate that could contribute to seizure relapse after unilateral surgery.”

Reviewer #3 (Remarks to the Author):

Thank you for inviting me to review this paper by Xie et al. This paper has several positive elements, including a novel functional-structural brain and normative framework in one of the largest temporal lobe epilepsy cohorts to date. The machine learning model is also constructed well in a nested approach. Despite these strengths of this work, I have several concerns (outlined below) that I hope the authors will find helpful:

We thank the Reviewer for the thoughtful evaluation of our work and for highlighting its strengths. We appreciate the feedback and address each point in detail below.

1) The title suggests a personalised biomarker of temporal lobe epilepsy in a cortical parcellation mask: Despite the scientific advance in this work, I am not sure the ‘personalised biomarker’ tag in the title is warranted based on the result. One of the reasons for this is that large amounts of the subcortical brain is not being analysed in this approach with the cortical Glasser 360 parcellation mask (<https://www.nature.com/articles/nature18933>). It would be great to hear why the subcortical brain was excluded from this work, particularly in a neurological condition where many people have a seizure focus from the mesial temporal lobes. Several subcortical masks are now available that may be convolved with the Glasser mask for full brain coverage: e.g., <https://elifesciences.org/articles/59430> & <https://pubmed.ncbi.nlm.nih.gov/32989295/> & <https://www.researchsquare.com/article/rs-4203104/v1>. Secondly, it is unlikely that machine learning results with AUC of 0.63-0.76 are strong enough to progress to individual biomarkers without further development.

We thank the Reviewer for these valuable suggestions. In response, we revised our analyses to include key subcortical structures and the hippocampus. The updated results (see revised **Figures 2, 3**) reveal significant functional deviations, particularly in the hippocampus, caudate, and putamen, underscoring the involvement of distributed cortico-subcortical networks in TLE. These findings have been incorporated into the revised **Results and Discussion (p. 6–8)**.

“As for the subcortex, the median proportion of extreme deviations was 6.0% for SV (range = 1.8%–9.2%), 8.9% for ReHo (4.6%–13.5%), and 3.7% for NS (1.8%–6.7%).”

“Next, aggregating the three metrics using a multivariate Mahalanobis distance revealed a diffuse spatial pattern of functional deviations across the brain in the TLE group (neocortex: median = 5.3% [1.8%–12.4%], subcortex: median = 5.9 [1.8%–9.9%], **Figure 2B**), with the highest overlap observed in the ipsilateral mesiotemporal lobe (11.4%; **Figure S3**), putamen (9.9%), and hippocampus (7.1%).”

“As for subcortical structures, bilateral hippocampi, pallidum and nucleus accumbens, together with ipsilateral caudate emerged as dominant epicenters (**Figure 3D**).”

The revised **Discussion** now clarified that our approach/biomarker is not intended to replace established diagnostic tools, but that it may serve as a supplemental aid within a multimodal framework. We also emphasized the need for further methodological refinement and integration with established clinical workflows before such approaches can be considered for translation to clinical practice. See **(p. 14)**:

“Diagnosis and preoperative evaluation of pharmacoresistant epilepsy is most effective when conducted through a multidisciplinary approach,⁸³ combining clinical, electrophysiological, and neuroimaging data. Our machine learning models, informed by individualized functional deviations, predicted postsurgical seizure outcome with moderate accuracy. While not sufficient for independent clinical decision-making, such biomarkers may offer complementary value when interpreted alongside established tools, including structural MRI, PET, electroclinical assessments, and intracranial EEG in more complex cases. As the field moves toward more personalized, network-based models of epilepsy, integrating functional MRI-derived metrics into multimodal frameworks may enhance outcome prediction

and improve surgical candidate stratification. Nonetheless, further work is needed to refine and validate these methods—particularly through prospective studies and harmonized protocols—before they can be reliably translated into clinical practice.”

2) Defining local, regional and global connectivity: Determining the spatial extent of correlations is challenging, especially when averaged across a larger brain area in a parcellation framework. Given its cubical constraints, I see the rationale for calling ReHo a regional measure, but NS is challenging. It would be good to see an Euclidean distance measure for NS to confirm that we are looking at global changes in the brain rather than reflecting correlations within spatially proximate nodes. Also, signal variability and power spectral density are intrinsically related, so it would be great to hear how the SV algorithm correlates with ALFF/fALFF (<https://pmc.ncbi.nlm.nih.gov/articles/PMC3902859/>)

We agree that the spatial scale of correlations is an important consideration, particularly in the context of parcellation-based analyses. As suggested, we examined how Euclidean distance modulates node strength (NS) deviations. We partitioned inter-regional connections into short-range (≤ 75 mm) and long-range (> 75 mm) groups (the cutoff distance is typically 75 mm)^{1,2,3} and recomputed NS within each set. TLE patients exhibited comparable deviation patterns for short-range and long-range NS. Notably, the long-range NS deviation pattern closely matched the full NS deviation map ($\rho = 0.92$), indicating that NS captures global connectivity alterations that are not solely driven by spatially local correlations. These findings are reported the in the revised **Results (p. 6)** and the new **Figure S2 (p. 6)**:

“To further explore the influence of connection distance (*i.e.*, Euclidean distance) on NS deviations, we computed short-range (≤ 75 mm) and long-range (> 75 mm) NS following previous work.³⁸⁻⁴⁰ TLE patients showed comparable deviation patterns for both short-range NS (median = 6.7%, range = 2.1%–11.7%; short-range vs. full NS: $\rho = 0.54$, $P_{\text{spin}} < 0.001$) and long-range NS (median = 6.4%, range = 2.1%–12.1%; long-range vs. full NS: $\rho = 0.92$, $P_{\text{spin}} < 0.001$; **Figure S2**). Notably, short-range NS deviations were more pronounced in the bilateral lateral prefrontal and inferior temporal cortices, while long-range NS deviations were greater in the ipsilateral insular cortex.”

Figure S2: Region-specific prevalence of extreme deviations in TLE patients in short-range (A) and long-range NS (B). (i) Proportion of patients with extreme deviations ($|W\text{-score}| \geq 1.96$) in each brain region. (ii) Mean proportion of extreme deviations in each lobe. (iii) Spatial correlations between full-range NS deviations (x-axis, **Figure 2A**) and short-range and long-range NS deviations (y-axis), with each dot representing a single brain region. Statistical significance (*i.e.*,

P_{spin}) of correlations is assessed using spin permutation tests with 5,000 iterations. NS = node strength; ipsi = ipsilateral; contra = contralateral; F = frontal; P = parietal; T = temporal; O = occipital; I = insula

1. He, Y. et al. Small-world anatomical networks in the human brain revealed by cortical thickness from MRI. *Cereb. Cortex* **17**, 2407-2419 (2007).
2. Liang, X. et al. Coupling of functional connectivity and regional cerebral blood flow reveals a physiological basis for network hubs of the human brain. *Proc. Natl. Acad. Sci. U.S.A.* **110**, 1929-1934 (2013).
3. Lv, H. et al. Resting-state functional MRI: Everything that nonexperts have always wanted to know. *AJNR Am. J. Neuroradiol.* **39**, 1390 (2018).

We also calculated regional deviations in the fractional amplitude of low-frequency fluctuations (fALFF; 0.01-0.08 Hz). Consistent with the signal variability results, we found marked deviations in TLE patients in lateral temporal, prefrontal and parietal cortices, and precuneus and posterior cingulate gyrus, as well as in the hippocampus and caudate, indicating the similarity between fALFF and signal variability metrics.

3) Multisite data without scanner harmonization? I found no information about applying between-scanner harmonization correction in these cohorts. Large amount of MRI research suggests that systematic scanner effects are an issue, but can be (satisfactory) corrected with approaches such as ComBat (<https://pmc.ncbi.nlm.nih.gov/articles/PMC6179920/>). Recent research indicates that large fMRI variability between scanners for the same subjects (https://direct.mit.edu/imag/article/doi/10.1162/imag_a_00042/118214/A-resource-for-development-and-comparison-of), so this is likely to be an issue especially for the machine learning analysis as two sites have controls and four sites have epilepsy subjects. In other words, there is an imbalance of groups between scanners.

To correct for variability related to scanners/sites, we applied ComBat harmonization to each rs-fMRI metric independently, using dataset as the batch variable. The ComBat model included age, sex (male/female), and diagnosis group (controls, TLE, extra-TLE) as covariates to preserve relevant inter-individual and group-level variability. This approach allows harmonization across datasets while maintaining variance attributable to key clinical and demographic factors, even when some diagnostic subgroups (*e.g.*, extra-TLE patients) were not represented in all sites. We clarified this point in the revised **Methods (p. 16)**:

“To reduce potential biases and non-biological variability induced by sites/scanners, we used ComBat to harmonize each rs-fMRI metric independently across participants.⁹⁷ ComBat uses multivariate linear mixed-effects regression and empirical Bayes to correct for batch effects.⁹⁸ It has previously been applied to eliminate site effects in multi-site rs-fMRI studies.^{99,100} In this work, dataset was treated as a batch variable, while preserving variances of interest,

including age, sex (males, females), and diagnosis (controls, TLE, extra-TLE). The harmonized metrics, corrected for inter-site variations, were subsequently used to generate W -score maps.”

We also include a discussion of the importance of harmonization in the revised **Discussion (p. 12)**:

“Encouragingly, we found that patterns of functional brain deviations were highly consistent across sites and robust to the inclusion of global signal regression as an additional denoising step. A key strength of our study lies in the harmonization of preprocessing and analytic workflows across centers, alongside statistical adjustment for site effects. This step enhances the reliability and generalizability of our multi-site findings by minimizing nonbiological variability arising from scanner and acquisition protocol differences.⁵⁸”

4) Smoothing, filtering and regressors:

- What filtering band was used for the fMRI connectivity data?

We applied temporal band-pass filtering in the 0.01–0.08 Hz to the fMRI connectivity data. This has been clarified in the revised **Supplementary Materials (p. 3)**:

“Resting-state fMRI preprocessing included discarding the first five volumes, reorientation, slice-timing correction, correction for head motion and distortion, and temporal band-pass filtering (0.01–0.08 Hz).”

- What was the rationale for using a large smoothing parameter extent (10mm) in this cohort?

We applied a 10 mm FWHM Gaussian kernel at the vertex-wise level, consistent with our prior work on cortical morphometry, microstructure, and functional connectivity in TLE.¹⁻⁴ This level of smoothing improves signal-to-noise ratio, facilitates inter-subject anatomical alignment on the surface, and better fulfills the assumptions of Gaussian random field theory used for cluster-level statistical inference. Importantly, surface-based kernels operate along the folded cortical sheet and are anatomically specific—constraining averaging within grey matter—whereas voxel-based isotropic kernels can blur across tissue boundaries and sulci, increasing partial-volume effects.⁵ Given the substantial heterogeneity in cortical morphology and functional organization in our clinical cohort, a 10mm kernel provided a pragmatic balance between sensitivity to spatially extended effects and regional specificity.

We clarified this point in the revised **Supplementary Materials (p. 3)**:

“Surface-based maps, including cortical thickness, FA, MD, and resting-state fMRI time series, were smoothed along the cortical sheet with a 10-mm FWHM Gaussian kernel. This surface-constrained smoothing improves signal-to-noise ratio, decreases variability, and retains sensitivity while limiting partial-volume blurring across tissue boundaries.^{1,9,13,14}”

1. Royer, J. et al. Cortical microstructural gradients capture memory network reorganization in temporal lobe epilepsy. *Brain* **146**, 3923-3937 (2023).
2. Xie, K. et al. Atypical connectome topography and signal flow in temporal lobe epilepsy. *Prog. Neurobiol.* **236**, 102604 (2024).
3. Xie, K. et al. Temporal lobe epilepsy perturbs the brain-wide excitation-inhibition balance: associations with microcircuit organization, clinical parameters, and cognitive dysfunction. *Adv. Sci.* **12**, e2406835 (2025).
4. Ngo, A. et al. Associations of cerebral blood flow patterns with gray and white matter structure in patients with temporal lobe epilepsy. *Neurology* **103**, e209528 (2024).
5. Lerch, J. P. & Evans, A. C. Cortical thickness analysis examined through power analysis and a population simulation. *NeuroImage* **24**, 163-173 (2005).

- Finally, how many parameters were included in the motion correction (on page 3 of supplementary materials)?

Motion correction employed a rigid-body 6-parameter model (3 translations and 3 rotations) along with motion spike regression. We clarified this point in the revised **Supplementary Materials (p. 3)**:

“Resting-state fMRI preprocessing included discarding the first five volumes, reorientation, slice-timing correction, correction for head motion and distortion, and temporal band-pass filtering (0.01–0.08 Hz). Motion correction was performed using a rigid-body model with 6 parameters by registering all timepoint volumes to the mean volume. Motion outlier volumes (spikes) were discarded using FSL’s motion outlier detection outputs.¹ To maintain the temporal continuity of timeseries, we subsequently filled these censored frames using a linear interpolation. Nuisance signals removal was conducted either using an in-house trained ICA-FIX classifier (for *MICs*),¹² or through regression of white matter and cerebrospinal fluid signals for other datasets.”

A. demographic information

(i) age distribution

(ii) TLE seizure focus

(iii) hippocampal sclerosis

B. local-to-global functional metrics

C. calculate individual deviations (W -score) per region (e.g., NS)

(i) build normative models

(ii) calculate deviations (W -score)

(iii) identify extreme deviations

Figure 1: Overview of participants and analysis pipeline. (A) Demographic characteristics of the healthy control ($n = 298$) and TLE groups ($n = 282$). (i) Age distributions. Boxes represent the interquartile range (IQR), with the lower and upper boundaries corresponding to the 25th and 75th percentiles. Whiskers extend to the minimum and maximum values within $1.5 \times \text{IQR}$ from the 25th and 75th percentiles. Each dot represents an individual participant. (ii) Proportion of TLE patients with left- or right-sided seizure focus in each dataset. (iii) Proportion of TLE patients with or without ipsilateral hippocampal sclerosis in each dataset. (B) Overview of methodology for calculating functional metrics from resting-state functional MRI (rs-fMRI) time series in each cortical region and subcortical structure: signal temporal variability, regional homogeneity, and node strength. (C) Schematic of individual functional deviation (W -score) estimation, with an example here for node strength (NS). (i) Building normative models in healthy individuals. Age- and sex-related variations in each rs-fMRI metric are modeled using a linear regression model in the healthy control group, yielding beta maps for intercept (β_0), age (β_1), sex (β_2), and standard deviation of residuals in each brain structure. (ii) Estimating deviations in patients. Predicted value for a given patient's age and sex (e.g., NS_{pred}) is calculated as $\beta_0 + \beta_1 \times \text{age} + \beta_2 \times \text{sex}$. W -scores are defined as the normalized deviation of the observed values from the corresponding predicted values. (iii) Identifying extreme deviations. Brain regions with W -scores exceeding ± 1.96 (i.e., $|W| \geq 1.96$) were classified as showing extreme deviations, corresponding to the upper and lower 2.5% of the normative distribution.

A. prevalence of extreme deviations in TLE (univariate)

(i) signal variability (SV)

(ii) regional homogeneity (ReHo)

(iii) node strength (NS)

B. prevalence of extreme deviations in TLE (multivariate)

Figure 2: Region-specific prevalence of extreme functional deviations in TLE patients. (A) Proportion of TLE patients with extreme deviations ($|W\text{-score}| \geq 1.96$) in each cortical region and subcortical structure for (i) signal variability (SV), (ii) regional homogeneity (ReHo), and (iii) node strength (NS). Spider plots show the mean proportion of extreme deviations in each cortical lobe defined on the HCPMMP1.0 atlas and in the subcortex. (B) (i) Proportion of TLE patients with multivariate W -scores (aggregating SV, ReHo and NS) exceeding ± 1.96 . (ii) Region-specific deviation prevalence distribution. Each dot represents an individual brain region, with its position indicating the prevalence values of the three metrics and its color denoting the composite deviation prevalence. (iii) Mean proportion of extreme deviations per lobe. (iv) Distribution of the number of extreme deviations per patient. (v) Proportion of TLE patients with extreme deviations in each subcortical structure. ipsi = ipsilateral; contra = contralateral.

A. SC-informed functional deviations

(i) group-level SC connectome

(ii) neighbor deviations

(iii) node-neighbor correlation

B. epicenter identification

C. SC-informed cortical epicenters

D. SC-informed subcortical epicenters

Figure 3: Network-based spreading of regional functional deviations. (A) (i) Group-level structural connectivity (SC) matrix from diffusion MRI of 100 unrelated healthy individuals. (ii) Schematic of functional deviations of a node (p_i) and its neighbors (\bar{p}_i). If the regional deviation depends on SC network organization, nodes connected to highly abnormal neighbors (*i.e.*, high prevalence) will be more likely to be affected, whereas nodes connected to healthy neighbors (*i.e.*, low prevalence) will be less likely to be affected. (iii) The functional deviation of a node (x-axis) is positively correlated with the mean deviation of neighbors (y-axis) to which they are structurally connected. Each dot represents a brain region. Statistical significance (*i.e.*, P_{spin}) of the correlation coefficient is assessed using spin permutation tests (5,000 iterations). (B) Schematic of disease epicenter identification. A node whose SC profile spatially strongly relates with the TLE-related functional deviation map in **Figure 2B** is considered a disease “epicenter”. Epicenter likelihood is defined as the Spearman correlation coefficient between two spatial maps. (C, D) SC-informed epicenter likelihood map. Statistical significance of the likelihood is determined using spin permutation tests (5,000 iterations, $P_{\text{spin}} < 0.05$). ipsi = ipsilateral; contra = contralateral.

RESPONSE TO REVIEWERS (NCOMMS-25-33479A)

We sincerely thank the Editors and Reviewers for their careful evaluation of our revision and for the constructive guidance that improved the clarity and rigor of the work. We are grateful that our revisions have resolved all concerns of all Reviewers, with no additional changes requested.

Reviewer #1 (Remarks to the Author):

No additional comments. The authors have satisfactorily addressed all my comments.

No additional changes were requested.

Reviewer #2 (Remarks to the Author):

The authors have comprehensively addressed my questions about the manuscript. I have no further questions or concerns.

No additional changes were requested.

Reviewer #3 (Remarks to the Author):

The authors have addressed my previous concerns. This is great work with impact in the field.

No additional changes were requested.